# Localized dopant motion across the 2D Ising phase transition

**Kristian Knakkergaard Nielsen**[1*]

**1** Max-Planck-Institut für Quantentoptik, D-85748 Garching, Germany

* kristian.knakkergaard.nielsen@mpq.mpg.de

## Abstract

I investigate the motion of a single hole in 2D spin lattices with square and triangular geometries. While the spins have nearest neighbor Ising spin couplings $J$, the hole is allowed to move only in 1D along a single line in the 2D lattice with nearest neighbor hopping amplitude $t$. The non-equilibrium hole dynamics is initialized by suddenly removing a single spin from the thermal Ising spin lattice. I find that for any nonzero spin coupling and temperature, the hole is *localized*. This is an extension of the thermally induced localization phenomenon [1] to the case, where there is a phase transition to a long-range ordered ferromagnetic phase. The dynamics depends only on the ratio of the temperature to the spin coupling, $k_B T/|J|$, and on the ratio of the spin coupling to the hopping $J/t$. I characterize these dependencies in great detail. In particular, I find universal behavior at high temperatures, common features for the square and triangular lattices across the Curie temperatures for ferromagnetic interactions, and highly distinct behaviors for the two geometries in the presence of antiferromagnetic interactions due geometric frustration in the triangular lattice.

# 1 Introduction

Is an impurity or a dopant necessarily delocalized in a system with polarized long-range order? As the system is almost perfectly homogeneous, the immediate response would presumably be 'yes'! However, if the dopant's motion is correlated with the background in which it moves, then it modifies the background as it moves. Therefore, this question is more subtle than one would immediately expect. Such a scenario arises in its most simplistic form in an Ising magnet with a doped hole. Here, the ability of the spins to hop only onto vacant sites, means that the motion of holes is completely contingent on a counterpropagating spin. While the hole in a ferromagnetic ground state will certainly delocalize, the same may not remain true as soon as one heats up the system by any infinitesimal amount. The general scenario turns out to be tremendously complex to analyze, however. In particular, the exponential growth in the number of possible configurations of the system as the hole moves away from its origin means that analytical treatments are generically out of the window. However, numerical investigations of full two-dimensional (2D) motion of holes in thermal spin ensembles have been carried out [2–4]. These intriguing results are unfortunately limited by the underlying exponential complexity of the dynamics in a generic spin state to short times and/or small systems, and no robust conclusions have been found for long timescales in the thermodynamic limit. A mean-field approach [5] has, however, shown the possibility of rich phase diagram, supporting stripe formation [6] at low temperatures.

In a broader context, dopants in magnetic lattices have been extensively studied for decades [7–11] due to their intimate links to high-temperature superconductivity [12–14]. In recent years, this line of research has seen a fruitful revival thanks to advances in quantum simulation experiments with ultracold atoms in optical lattices [15–22]. Such experiments can also manipulate the models otherwise fixed in the solid state. In particular, clear signatures of magnetically mediated pairing of dopants [22] has been observed and boosted by only allowing holes to move in one dimension (1D)*along* the investigated ladder geometry. Building on these successes, I recently investigated such a model idealised further to support only Ising type spin couplings [1]. Contrary to earlier setups [2–4], 1D motion in the an Ising magnet facilitates the investigation of very large system sizes and essentially arbitrarily long evolution times can be achieved. This allowed me to systematically show that the hole is *localized* for any nonzero temperature and any nonzero spin coupling, and only asymptotically delocalize in the limits $\beta J = J/k_B T \rightarrow -\infty$ and/or $|J|/t \rightarrow 0$. In other words, even though there are perfect quasiparticle excitations in these limits, in which they behave exactly as free particles, any nonzero temperature and spin coupling immediately localizes the hole.

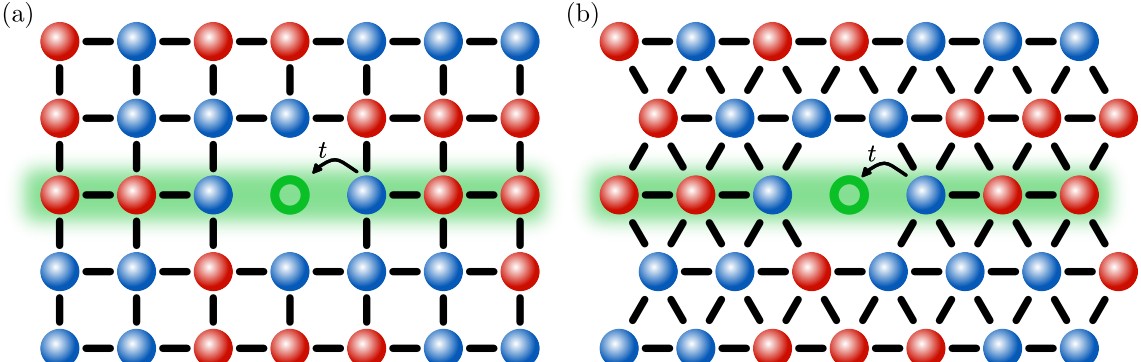

Figure 1: The mixed-dimensional $t$-$J_z$ model with 1D dopant motion (green region) in a square (a) and triangular (b) spin lattice. Here, both spin-$\uparrow$ (red spheres) and $-\downarrow$ particles (blue spheres) may hop to vacant sites (green circles) – holes – with hopping amplitude $t$ *along* the system. The spins are coupled via isotropic Ising interactions with magnitude $J$ (black lines).

From a statistical mechanics perspective, the ladder geometry is, however, a one-dimensional system. Consequently, there is no phase transition for the underlying Ising spins until zero temperature is reached. One might think, therefore, that the hole localization is linked directly to the disordered nature of the finite temperature (Gibbs) state. This is supported in my previous paper by the fact that the localization length becomes proportional to the spin-spin correlation length at low temperatures. Since this length scale diverges as the para- to ferromagnetic phase transition is crossed from above, this would seem to indicate that for a fully 2D spin lattice, the hole should delocalize as the transition to a ferromagnetic phase is crossed at the Curie temperature. As a result, the present Article is concerned with addressing exactly this question: how is the 1D motion of the hole affected in the presence of the Ising phase transition for the 2D Ising model? To study this carefully, I will consider two lattice geometries: the square and triangular lattices with nearest neighbor Ising interactions, see Fig. 1. These support such a para- to ferromagnetic phase transition at their respective Curie temperatures. The comparison of the two lattice structures allow me to extract the common features of the systems and highlight the importance of the change in the number of nearest neighbor spin couplings (from 4 for the square lattice to 6 in the triangular case). To my surprise, the realization of a long-range ordered ferromagnetic phase is *not* enough to delocalize the hole. Instead, I find that the exponentially small probability of meeting spin flips in the ferromagnetic phase is more than adequate to keep the hole localized.

The localization effect in the two-leg ladder was tied directly to thermal spin fluctuations of the disordered spin lattice. This realizes a novel variant of Anderson localization in the presence of *strong* disorder, in which the dopant back-scatter off the spin fluctuations, as the energy cost of propagating further away from its origin will inevitably fluctuate to values larger than the initial kinetic energy of the hole. This framework also explains the localization quantitatively well in the disordered phase above the Curie temperature in this present setup. However, in the ordered phase this picture breaks down for intermediate to large values of $|J|/t$. In particular, the thermal spin fluctuations in the long-range ordered phase on short length scales tend to happen as singular spin flips. This results in a crossover between large and small values of $|J|/t$. For large $|J|/t$, the hole backscatters off these single spin flips. For intermediate values, it tunnels through many such flips, but destructively interfere for different pathways to a specific point because there is a large statistical variation in how long the segments are between such spin flips. This is the physics of Anderson localization in the presence of *weak* disorder [23]. Finally, for sufficiently small $|J|/t$ this effect is too weak. Instead the hole once again backreflects once the build up of potential energy overcomes its initial kinetic energy.

Moreover, for a square lattice its bipartite structure results in a symmetry between the ferro- and antiferromagnetic case, such that the thermodynamics of the underlying spin systems are equivalent. Therefore, the Neél critical temperature is simply the same as the Curie temperature for ferromagnetic couplings. However, the motion of the hole is markedly different in the two scenarios. In the antiferromagnet, the buildup of the staggered Neél order leads to a linearly increasing potential which becomes stronger and stronger as zero temperature is approached. In this case, there is, therefore, a crossover between localization driven by spin fluctuations at high temperatures to localization exerted by an effective confining potential at low temperatures. The latter eventually leads to heavy coherent oscillations in the position of the hole due to interference of the low-energy states as zero temperature is approached, analogous to what was found for the two-leg ladder at zero temperature [24].

In stark constrast, in the triangular case, antiferromagnetic couplings hinder any phase transition all the way down to zero temperature and results in a residual entropy $S_0/N \simeq 0.323$ of the ground state manifold, due to an exponentially large ground state degeneracy [25,26]. The present setup, hereby, also allows me to study the influence of this extensive ground state degeneracy. We will see that while the behavior for the square antiferromagnet is characterized by more and more coherent oscillations of the hole's motion, the same is *not* true for the triangular antiferromagnetic case, and the dynamics retains a thermal character with smooth and incoherent behavior, even as zero temperature is approached.

From a traditional condensed matter point of view, it is hard to imagine how one would actually investigate the dynamical phenomena detailed above, as it requires one to track the motion of dopants in real time. However, quantum simulation experiments with ultracold atoms have made substantial breakthroughs in this regard. Not only do these systems allow for single site detection [27,28], but has enough sensitivity to actually track the motion of holes in real time [16]. Moreover, the Ising type of interactions investigated are e.g. facilitated with Rydberg-dressed atoms in optical lattices [29], which crucially still allow for the motion of dopants. Finally, the 1D restriction of the hole motion have been achieved for dopants in such spin lattices [22]. In this manner, one can combine these well-established experimental capabilities to observe the predicted effects, as I have also previously pointed out in a suggestion for an explicit experimental protocol [1].

The Article is organized as follows. In Sec. 2, the system and setup is explained. In Sec. 3, I describe the employed Monte-Carlo simulation of exact trajectories. In Sec. 4, a detailed analysis of the numerical results is given at infinite temperature [Sec. 4.1], for ferromagnetic [Sec. 4.2] and antiferromagnetic couplings [Sec. 4.2], and finally the general spin-coupling dependency [Sec. 4.4]. Moreover, a detailed discussion on the extension of the model is given in Sec. 5, before I conclude in Sec. 6.

## 2   System and setup

In this paper, I will consider the mixed-dimensional $t$-$J_z$ model

$$\hat{H} = \hat{H}_t + \hat{H}_J = -t \sum_{\langle \mathbf{i,j} \rangle_\parallel, \sigma} \left[ \tilde{c}^\dagger_{\mathbf{i},\sigma} \tilde{c}_{\mathbf{j},\sigma} + \text{h.c.} \right] + J \sum_{\langle \mathbf{i,j} \rangle} S^{(z)}_{\mathbf{i}} S^{(z)}_{\mathbf{j}}, \tag{1}$$

in the presence of a single dopant – a hole. Here, the correlated motion of the dopant is allowed through the nearest neighbor hopping Hamiltonian $\hat{H}_t$, with constrained operators $\tilde{c}^\dagger_{\mathbf{i},\sigma} = \hat{c}^\dagger_{\mathbf{i},\sigma}(1 - \hat{n}_{\mathbf{i}\bar\sigma})$ to ensure at most a single particle per site. Note that $\bar\sigma$ designates the opposite spin of $\sigma$, i.e. $\bar\uparrow = \downarrow$ and vice versa. Also, while the Ising spin couplings $\hat{H}_J$ couple all nearest neighbors isotropically, the hopping is *only* allowed along a 1D line, indicated by $\langle \mathbf{i,j} \rangle_\parallel$.

In a recent paper [1], I studied this model in a two-leg square ladder system. Here, I will extend the studies to a 2D spin lattice and encompass both square and triangular lattices. The main idea is to study the importance of the appearing Ising phase transition on the motion of the single hole, as well as understanding the importance of magnetic frustration in the triangular case for antiferromagnetic couplings.

To easily describe hole and spin degrees of freedom, I employ an exact Holstein-Primakoff transformation with the ferromagnetic state $|\text{FM}\rangle = |\cdots \uparrow\uparrow \cdots\rangle$ with all spins pointin up as the reference state. This leads to the alternate expressions for the hopping

$$\hat{H}_t = t \sum_{\langle \mathbf{i},\mathbf{j}\rangle_\parallel} \Big[ \hat{h}_\mathbf{j}^\dagger F(\hat{h}_\mathbf{i},\hat{s}_\mathbf{i}) F(\hat{h}_\mathbf{j},\hat{s}_\mathbf{j})\hat{h}_\mathbf{i} + \hat{h}_\mathbf{j}^\dagger \hat{s}_\mathbf{i}^\dagger F(\hat{h}_\mathbf{i},\hat{s}_\mathbf{i}) F(\hat{h}_\mathbf{j},\hat{s}_\mathbf{j})\hat{s}_\mathbf{j} \hat{h}_\mathbf{i} \Big] + \text{H.c.}, \tag{2}$$

and the spin coupling Hamiltonians

$$\hat{H}_J = J \sum_{\langle \mathbf{i},\mathbf{j}\rangle} \Big[ \frac{1}{2} - \hat{s}_\mathbf{i}^\dagger \hat{s}_\mathbf{i} \Big]\Big[ \frac{1}{2} - \hat{s}_\mathbf{j}^\dagger \hat{s}_\mathbf{j} \Big]\Big[ 1 - \hat{h}_\mathbf{i}^\dagger \hat{h}_\mathbf{i} \Big]\Big[ 1 - \hat{h}_\mathbf{j}^\dagger \hat{h}_\mathbf{j} \Big]. \tag{3}$$

Here, the spin excitation operator $\hat{s}_\mathbf{i}^\dagger$ is bosonic, and creates a spin-$\downarrow$ on site $\mathbf{i}$. Also, the hole is created by the operator $\hat{h}_\mathbf{i}^\dagger$, and maintains the statistics of the underlying spins, be it fermionic *or* bosonic [24]. In the hopping Hamiltonian $\hat{H}_t$, the operator $F(\hat{h},\hat{s}) = \sqrt{1 - \hat{s}^\dagger\hat{s} - \hat{h}^\dagger\hat{h}}$ keeps the single-occupancy constraint in check. This construction enables me to succinctly describe the motion of holes.

## 3 Monte Carlo sampling of exact trajectories

The non-equilibrium dynamics of the holes is initialized in the following manner. The system starts out in the absence of holes in its thermal Gibs state, $\hat{\rho}_J = e^{-\beta \hat{H}_J}/Z_J$. I assume nothing about how this equilibrium is initially established, but from the time of the quenched insertion of the hole at $\tau = 0$, I assume the system to be closed. The state of the system immediately after the removal of a spin from the origin $\mathbf{i} = \mathbf{0}$ is

$$\hat{\rho}(\tau = 0) = \sum_{\sigma_0} \hat{c}_{\mathbf{0},\sigma_0} \hat{\rho}_J \hat{c}_{\mathbf{0},\sigma_0}^\dagger = \hat{h}_\mathbf{0}^\dagger \hat{\rho}_J \hat{h}_\mathbf{0} + \hat{h}_\mathbf{0}^\dagger \hat{s}_\mathbf{0} \hat{\rho}_J \hat{s}_\mathbf{0}^\dagger \hat{h}_\mathbf{0} \tag{4}$$

After that, since the system is assumed to be closed, it evolves unitarily under the full Hamiltonian $\hat{H}$ in Eq. (1), $\hat{\rho}(\tau) = e^{-i\hat{H}\tau}\hat{\rho}(0)e^{+i\hat{H}\tau}$. Next, I express the density operator in the Ising basis with spin configurations $\boldsymbol{\sigma}$. Using this, I can write the time-evolved density matrix as the Boltzmann-weighted sum of pure-state evolutions

$$\hat{\rho}(\tau) = \sum_{\sigma_0,\boldsymbol{\sigma}} \frac{e^{-\beta E_J(\sigma_0,\boldsymbol{\sigma})}}{Z_0} |\Psi_{\boldsymbol{\sigma}}(\tau)\rangle \langle \Psi_{\boldsymbol{\sigma}}(\tau)|, \tag{5}$$

where $E_J(\sigma_0,\boldsymbol{\sigma})$ is the magnetic energy of the spin realization $\sigma_0,\boldsymbol{\sigma}$ before the hole is introduced. With the hole and spin excitation operators at hand, we may express the non-equilibrium pure states $|\Psi_{\boldsymbol{\sigma}}(\tau)\rangle$ quite concisely. In particular, I consider a system of size $(2N_x + 1) \times (2N_y + 1)$ with open boundary conditions, such that the coordinates is written as $\mathbf{i} = x,y$ with $x \in \{-N_x, -N_x + 1, \ldots, N_x\}$ and $y \in \{-N_y, -N_y + 1, \ldots, N_y\}$, and with the hole moving along the $y = 0$ leg of the 2D system. Here, the triangular lattice is implemented as a square lattice but with one additional spin coupling along one of the diagonals of the lattice [30].

In terms of the spin excitation operators, a certain subset $S_{\boldsymbol{\sigma}}^y$ in each leg $y$ will have spin flips. This means that the initial wave function for such a realization can be expressed as

$$|\Psi_{\boldsymbol{\sigma}}(\tau=0)\rangle = \hat{h}_{0,0}^\dagger \prod_{y=-N_y}^{N_y} \prod_{j\in S_{\boldsymbol{\sigma}}^y} \hat{s}_{j,y}^\dagger \, |\text{FM}\rangle. \tag{6}$$

When the hole moves along the system, the spins in leg $y=0$ countermove by a single lattice site, while all other spins remain static. Therefore, the state at any later time $\tau$ is

$$|\Psi_{\boldsymbol{\sigma}}(\tau)\rangle = \Bigg[ \sum_{x\geq 0} C_{\boldsymbol{\sigma}}(x,\tau) \hat{h}_{x,0}^\dagger \prod_{\substack{j\in S_{\boldsymbol{\sigma}}^0 \\ 0\leq j\leq x}} \hat{s}_{j-1,0}^\dagger \prod_{\substack{j\in S_{\boldsymbol{\sigma}}^0 \\ j>x}} \hat{s}_{j,0}^\dagger$$

$$+ \sum_{x<0} C_{\boldsymbol{\sigma}}(x,\tau) \hat{h}_{x,0}^\dagger \prod_{\substack{j\in S_{\boldsymbol{\sigma}}^0 \\ x\leq j<0}} \hat{s}_{j+1,0}^\dagger \prod_{\substack{j\in S_{\boldsymbol{\sigma}}^0 \\ j<x}} \hat{s}_{j,0}^\dagger \Bigg] \prod_{y\neq 0} \prod_{j\in S_{\boldsymbol{\sigma}}^y} \hat{s}_{j,y}^\dagger \, |\text{FM}\rangle. \tag{7}$$

The upper (lower) line describes the scenario in which the hole has moved $|x|$ sites to the right (left), and how the spin excitations countermove by one site to the left (right). Crucially, the probability amplitude to find the hole at site $x$ and time $\tau$ for a given spin realization $\boldsymbol{\sigma}$ only depends on these three variables, since the spin background is static apart from the countermotion due to the hole motion. This also means that the probability to observe the hole at position $x$ after time $\tau$ is the thermal average of $|C_{\boldsymbol{\sigma}}(x,\tau)|^2$,

$$P(x,\tau) = \text{tr}\Big[\hat{h}_{x,0}^\dagger \hat{h}_{x,0} \hat{\rho}(\tau)\Big] = \sum_{\sigma_0,\boldsymbol{\sigma}} \frac{e^{-\beta E_J(\sigma_0,\boldsymbol{\sigma})}}{Z_J} |C_{\boldsymbol{\sigma}}(x,\tau)|^2. \tag{8}$$

In this way, we have to determine the probability amplitudes $C_{\boldsymbol{\sigma}}(x,\tau)$ for a given spin realization $\boldsymbol{\sigma}$ and then perform sampling of the thermal average in Eq. (8). To determine $C_{\boldsymbol{\sigma}}(x,\tau)$, we may realize, in complete analogy to my previous paper on the two-leg ladder [1], that the $C_{\boldsymbol{\sigma}}(x,\tau)$ amplitudes obey free-particle equations of motion

$$i\partial_\tau C_{\boldsymbol{\sigma}}(x,\tau) = V_{\boldsymbol{\sigma}}(x)C_{\boldsymbol{\sigma}}(x,\tau) + t\left[C_{\boldsymbol{\sigma}}(x-1,\tau) + C_{\boldsymbol{\sigma}}(x+1,\tau)\right]. \tag{9}$$

with an emergent potential $V_{\sigma}(x)$. In particular, the potential may be divided into two parts: $V_{\boldsymbol{\sigma}} = V_{\boldsymbol{\sigma},\parallel} + V_{\boldsymbol{\sigma},\perp}$. The first term gives the contributions within the leg $y=0$ the hole is propagating in,

$$V_{\boldsymbol{\sigma},\parallel}(x) = J[\sigma_{1,0}\sigma_{-1,0} - \sigma_{x,0}\sigma_{x+1,0}], \; x>0,$$
$$V_{\boldsymbol{\sigma},\parallel}(x) = J[\sigma_{1,0}\sigma_{-1,0} - \sigma_{x,0}\sigma_{x-1,0}], \; x<0. \tag{10}$$

Here $\sigma_{x,y}=\pm 1/2$ designates spin-↑ $(+)$ and -↓ $(-)$ at site $\mathbf{i}=x,y$ of the original sample, i.e. *before* the hole has started to move. The second term $V_{\boldsymbol{\sigma},\perp}$ describes the trans-leg potential giving the contributions from the neighboring legs $y=\pm 1$. This depends on the geometry of the couplings. For the square lattice,

$$V_{\boldsymbol{\sigma},\perp}(x) = J \sum_{y=\pm 1} \sum_{j=+1}^{x} \sigma_{j,0}[\sigma_{j-1,y} - \sigma_{j,y}], \; x>0,$$

$$V_{\boldsymbol{\sigma},\perp}(x) = J \sum_{y=\pm 1} \sum_{j=-1}^{x} \sigma_{j,0}[\sigma_{j+1,y} - \sigma_{j,y}], \; x<0. \tag{11}$$

For the triangular lattice,

$$V_{\boldsymbol{\sigma},\perp}(x) = J \sum_{j=+1}^{x} \sigma_{j,0} \left\{ [\sigma_{j-1,+1} - \sigma_{j+1,+1}] + [\sigma_{j-2,-1} - \sigma_{j,-1}] \right\}, \; x > 0,$$

$$V_{\boldsymbol{\sigma},\perp}(x) = J \sum_{j=-1}^{x} \sigma_{j,0} \left\{ [\sigma_{j+2,+1} - \sigma_{j,+1}] + [\sigma_{j+1,-1} - \sigma_{j-1,-1}] \right\}, \; x < 0. \tag{12}$$

Here, the coordinates of the triangular lattices is defined by using the standard embedding on a square lattice with an additional diagonal coupling [25]. With these explicit constructions, the effective hole potential $V_{\boldsymbol{\sigma}}(x)$ can easily be computed from each realized spin sample $\boldsymbol{\sigma}$. Furthermore, for each $\boldsymbol{\sigma}$ Eq. (9) can be solved highly efficiently by defining the effective Hamiltonian $\mathcal{H}_{\boldsymbol{\sigma}}$ with diagonal entries given by the effective potential, $\mathcal{H}_{\boldsymbol{\sigma}}(x,x) = V_{\boldsymbol{\sigma}}(x)$, and off-diagonal entries given by the hopping $\mathcal{H}_{\boldsymbol{\sigma}}(x, x \pm 1) = t$. By concatenating $C_{\boldsymbol{\sigma}}(x, \tau)$ as a vector $\mathbf{C}_{\boldsymbol{\sigma}}(\tau)$, the effective Schrödinger equation

$$i \partial_\tau \mathbf{C}_{\boldsymbol{\sigma}}(\tau) = \mathcal{H}_{\boldsymbol{\sigma}} \mathbf{C}_{\boldsymbol{\sigma}}(\tau) \tag{13}$$

can be solved using standard linear algebra packages. As $\mathcal{H}_{\boldsymbol{\sigma}}$ is a sparse matrix, I use the "expm_multiply" function, part of the "scipy.sparse.linalg" package in Python. This allows me to go to systems sizes of at least 10.000 sites long.

To obtain the samples $\boldsymbol{\sigma}$ in the first place, I perform Monte-Carlo sampling. In the presence of a phase transition, i.e. for FM couplings in the triangular lattice as well as both FM and AFM couplings in the square lattice, I use a combined Wolff [31, 32] and Metropolis-Hastings [33, 34] algorithm to increase the accuracy around the critical temperature. For AFM couplings in the triangular lattice, there is no phase transition, and I instead simply use a standard Metropolis-Hasting algorithm with single-spin flip updates. These algorithms are used to generate 2000 samples for a range of inverse temperatures $\beta J$. In this manner, the dynamics of the hole is computed highly efficiently, whereby I easily go to very large system sizes, long evolution times and effortlessly monitor the behavior across the phase transition. The main observable in this regard will be the root-mean-square (rms) distance, calculated as

$$x_{\text{rms}}(\tau) = \left[ \sum_x x^2 P(x, \tau) \right]^{1/2}, \tag{14}$$

evaluated from the hole probability distribution function $P(x, \tau)$ in Eq. (8). This methodology is completely equivalent to the one I developed in Ref. [1]. Only the explicit expressions for the trans-leg potentials in Eqs. (11) and (12) differ, as well as the implementation of the Wolff algorithm. From the rms dynamics, I extract a localization length as the long-time average

$$l_{\text{loc}} = \lim_{\tau \to \infty} \frac{1}{\tau} \int_0^\tau ds \, x_{\text{rms}}(s). \tag{15}$$

As a good check of the sampling, I compute the average and variance of the effective hole potential. As for the two-leg ladder [1], these are both found to be linear in the distance $|x|$,

$$\langle V_{\boldsymbol{\sigma}}(x) \rangle = |J| \frac{|x|}{x_{\text{ave}}} + \text{const.}, \quad \text{Var}[V_{\boldsymbol{\sigma}}(x)] = J^2 \frac{|x|}{x_{\text{fl}}} + \text{const.}, \tag{16}$$

with temperature-dependent length scales $x_{\text{ave}}$ and $x_{\text{fl}}$, respectively. These are explicitly shown in Fig. 2. The former may also quite easily be expressed in terms of short-range correlators as

$$x_{\text{ave}}^{\square} = \frac{2}{C(1) - C(\sqrt{2})}, \quad x_{\text{ave}}^{\triangle} = \frac{2}{C(1) - C(\sqrt{13/4})}, \tag{17}$$

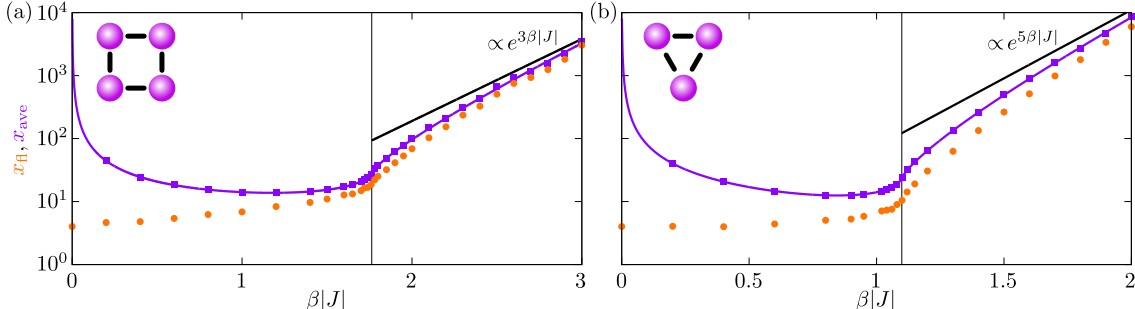

Figure 2: Length scales associated with the effective hole potential for the square (a) and triangular (b) lattices for ferromagnetic couplings, $J < 0$, as a function of the inverse temperature $\beta|J|$. The length scales are defined through the linearity of the mean and variance of the potential: $\langle V_{\sigma}(x)\rangle = |J| \cdot |x|/x_{\text{ave}}$ and $\text{Var}[V_{\sigma}(x)] = J^2 \cdot |x|/x_{\text{fl}}$, respectively. The solid violet lines show the analytic solutions in Eq. (17), while the black lines give the asymptotic behaviors, scaling as $e^{3\beta|J|}$ and $e^{5\beta|J|}$ for the square and triangular cases. The thin vertical black lines designate the position of the para- to ferromagnetic phase transition at $\beta_c|J| = 2\ln(1 + \sqrt{2})$ and $\beta_c|J| = \ln(3)$, respectively.

for the square (left) and triangular (right) lattices, respectively. The nearest-neighbor correlators $C(1) = 4\langle \hat{S}_{0,0}^{(z)} \hat{S}_{1,0}^{(z)}\rangle$, and next-nearest neighbor correlators $C(\sqrt{2}) = C(\sqrt{2}) = 4\langle \hat{S}_{0,0} \hat{S}_{1,1}\rangle$ and $C(\sqrt{13/4}) = 4\langle \hat{S}_{1,0} \hat{S}_{0,1}\rangle$ are computed explicitly in Appendices A and B and are seen to agree perfectly with the numerical results in Fig. 2.

## 4   Results

In this section, I present the results for the hole dynamics. The section is split into three subsections, describing the universal infinite temperature limit in Sec. 4.1, ferromagnetic couplings in Sec. 4.2, and finally antiferromagnetic couplings in Sec. 4.3.

### 4.1   Infinite temperatures

For infinite temperatures, $\beta J = 0$, the results are quite similar to the two-leg ladder case [1]. In particular, since each spin is now an independent random variable $\sigma = \pm 1/2$, it follows that the potential $V_{\sigma}(x)$ performs a random walk as a function of the distance $|x|$. In particular, the mean value of the potential over all the spin realization vanishes identically, while the standard deviation scales as $\sqrt{|x|}$,

$$\sigma[V_{\sigma}(x)] = \frac{J}{2}\sqrt{|x| + 1}. \tag{18}$$

This gives a length scale $x_{\text{fl}} = 4$ at infinite temperatures in excellent agreement with the Monte-Carlo sampling result shown in Fig. 2. As was also realized for the two-leg ladder [1], the dynamics in this limit becomes independent of the sign of $J$. This can be seen directly from the explicit expressions in Eqs. (10)-(12). Here, a sign change in $J$ can be absorbed in $\sigma_{j,0} = \pm 1/2$, as this random variable is independent from the rest of the spins at infinite temperatures. More interestingly, since the expressions for the square and triangular lattices, respectively Eqs. (11) and (12), have the same number of terms with the same quadratic structure, $\sigma\sigma'$, they give identical potentials *at infinite temperature*. Hence, the dynamics is not only independent of the sign of the spin coupling in this limit, but also whether the

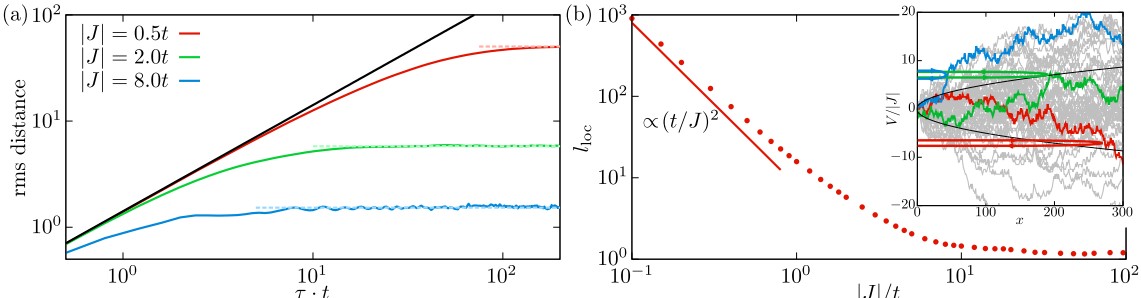

Figure 3: **Universal infinite temperature dynamics.** (a) Root-mean-square (rms) distance as a function of time $\tau$ in units of the hopping $t$ for indicated spin couplings at infinite temperature. The dashed lines are the asymptotic averages defining the localization length $l_{\mathrm{loc}}$. All lines collapse at short times to an initial ballistic motion with speed $\sqrt{2}t$ (black line). (b) Localization length $l_{\mathrm{loc}}$ at infinite temperature as a function of the spin coupling (red dots). For small $|J|/t$, the localization length diverges as $(t/J)^2$ (red solid line). Inset: effective hole potential $V(x)$ in units of the spin coupling for 50 spin samples (grey lines). As a hole with initial kinetic energy $\sim t$ travels in a specific spin sample (colored lines), it will eventually back-scatter off the potential (colored lines with arrows, shown for $t \simeq 5|J|$), because the standard deviation of the potential grows as $|J|\sqrt{|x|+1}/2$ (black lines), see also Eq. (18).

geometry is square or triangular. I note, however, that this does not hold for general 2D structures. Indeed, for kagome and honeycomb lattices, one can find 1D lines, in which the structure is the same as the two-leg ladder, and consequently gives the same hole motion along these lines as in the two-leg ladder at infinite temperatures. In Fig. 3(a), I plot the root-mean-square distance, calculated from Eq. (14), versus time for indicated values of the spin coupling. As was found for the two-leg ladder [1], the dynamics crosses over from an initial universal ballistic behavior [35] of a free particle [with speed $\sqrt{2}t$], to localized dynamics on long timescales. As the spin coupling is lowered, the associated localization length, shown in Fig. 3(b), increases and eventually diverges as $(t/J)^2$ in the limit of $|J|/t \ll 1$. As was realized for the two-leg ladder, the localization can be understood by equating the initial kinetic energy of the hole $\sim t$ to the fluctuations of the potential at a length scale $l_{\mathrm{fl}}$, i.e. the standard deviation $\sigma(V_{\boldsymbol{\sigma}}(l_{\mathrm{fl}})) \simeq |J|\sqrt{l_{\mathrm{fl}}}/2$. This results in the fluctuation-induced localization length,

$$l_{\mathrm{fl}} \simeq 4\left[\frac{t}{J}\right]^2, \tag{19}$$

which, apart from an overall factor of 2, is in quantitative agreement with the observed results in Fig. 3(b). The physical picture is, hereby, that the dopant will eventually back-scatter off the emergent effective potential $V(x)$ [inset of Fig. 3(b)]. Indeed, taking the enhancement factor of 2 into account, I check when $|V_{\boldsymbol{\sigma}}(x)|$ first exceeds $\sqrt{2}t$ for each spin realization $\boldsymbol{\sigma}$ and average over the achieved mean-square distance $x^2$ over all the spin realizations. This quantitatively recovers the full numerical solution both in terms of the localization length $l_{\mathrm{loc}}$, and in terms of the standard deviation around this value.

## 4.2 Ferromagnetic couplings

In this Section, I take a detailed look at the temperature dependency for ferromagnetic couplings. In Figs. 4(a) and 4(b), I show the rms dynamics across the phase transitions at the inverse Curie temperatures $\beta_c^{\square}|J| = 2\ln(1+\sqrt{2}) \simeq 1.76$ and $\beta_c^{\triangle}|J| = \ln(3) \simeq 1.10$, for the square and triangular cases respectively. At short times, they again all collapse to a ballistic expansion with speed $v = \sqrt{2}t$, as they should [35]. For lower temperatures – higher $\beta|J|$

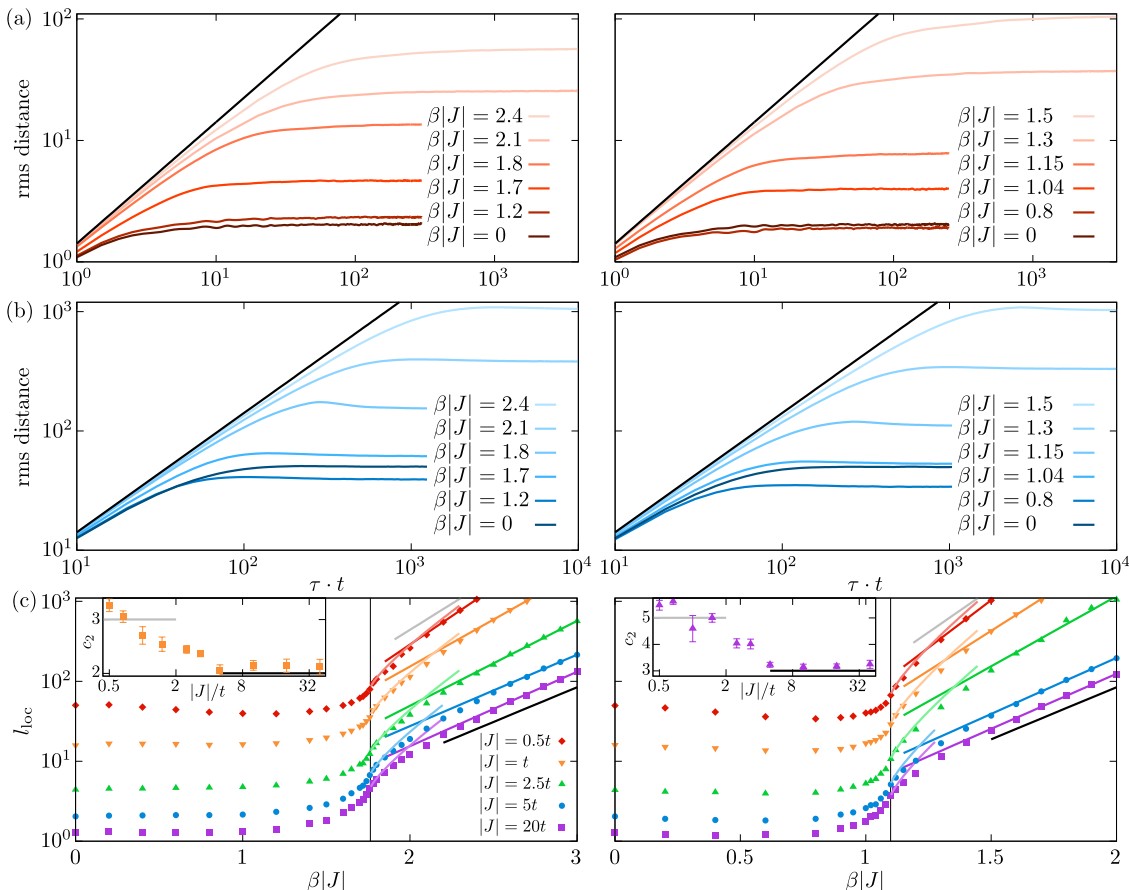

Figure 4: **Ferromagnetic couplings.** Root-mean-square distance (rms) dynamics for $J < 0$ and indicated inverse temperatures $\beta|J|$ in the square (left) and triangular (right) lattices for $|J| = 5t$ in (a) and $|J| = 0.5t$ in (b). At long times and any finite $\beta|J|$, the rms distance saturates at long times. The associated localization length is plotted in (c) versus inverse temperature for indicated values of the spin coupling for the square (left) and triangular (right) lattices. The vertical black lines indicate the phase transitions at $\beta_c^{\square}|J| = 2\ln(1+\sqrt{2})$ (left) and $\beta_c^{\triangle}|J| = \ln(3)$ (right). At low temperatures, the localization length scales as $c_1 \exp(c_2\beta|J|)$ (solid lines). The growth rate $c_2$ is extracted and plotted in the insets of (c) as a function of $|J|/t$. The horizontal black and grey lines show the expected limiting behaviors for small and large values of $|J|/t$, corresponding to the black and grey lines in the main panels.

– the dynamics generally follows this ballistic behavior for longer. Essentially, this is because the system becomes more and more ferromagnetically ordered, allowing the hole to move more freely. Note, however, that there are exceptions to this general rule. For example, for $|J| = 0.5t$, we see that the dynamics at $\beta|J| = 1.2$ in the square lattice and $\beta|J| = 0.8$ in the triangular lattice is *more* localized than at infinite temperature. We will return to this subtlety later on.

Moreover, it is strikingly apparent that the asymptotes, i.e. the localization lengths of the hole, remain *finite* even in the phase with long-range ferromagnetic order. This is surprising with the analysis of the two-leg ladder in mind [1]. Here, it was shown that the localization length scales with the spin-spin correlation length at low temperatures. As this length scale diverges across the para- to ferromagnetic phase transition in the present 2D system, the expectation from there would be that the hole should also delocalize across the transition.

To analyze this puzzling situation further, I next calculate the localization length across the

phase transitions for the square and triangular lattices in Fig. 4(c). This manifestly shows that even though the localization has a sharp increase around the phase transition, no divergence appears. Instead, I find that the localization length scales as

$$l_{\text{loc}} = c_1 \left( \frac{|J|}{t} \right) \exp \left[ c_2 \left( \frac{|J|}{t} \right) \beta |J| \right], \tag{20}$$

for low temperatures. Here, the coefficients $c_1$ and $c_2$ are functions of $|J|/t$. The exponential growth rate, $c_2$, is plotted in the insets of Fig. 4(c), and is seen to increase for decreasing $|J|/t$, between what seems to be two limiting behaviors. This is again in stark contrast to the two-leg ladder case [1]. Here, the localization length scales with the spin-spin correlation length at low temperatures, which in turn increases exponentially as $\exp[\beta|J|]$, i.e. with an exponential coefficient $c_2 = 1$ independent of $|J|/t$.

Let us, therefore, analyze these limits in detail. First, for a large mobility of the hole, $|J| \ll t$, we can use a semi-classical argument of energetic turning points. The basic idea is that the average and standard deviation of the effective hole potential defines two length scales which compete to localize the hole even as $|J|/t$ becomes small. In particular, I found in Fig. 2 that the mean and standard deviation at all temperatures and ferromagnetic couplings behave as $\langle V_\sigma(x) \rangle = |J||x|/x_{\text{ave}}$ and $\sigma(V_\sigma(x)) = |J|\sqrt{|x|/x_{\text{fl}}}$. By equating these to the initial kinetic energy of the hole $\sim t$, we obtain the semi-classical turning points

$$l_{\text{ave}} = \frac{t}{|J|} x_{\text{ave}}, \; l_{\text{fl}} = \left( \frac{t}{J} \right)^2 x_{\text{fl}}, \tag{21}$$

for the average and standard deviation of the potential, respectively. Whichever of these two length scales is the shortest is expected to describe the localization as $|J|/t$ becomes small – where it is well-defined to talk about an initial kinetic energy of the hole. This actually also explains the non-monotonic behavior of the localization length at high temperatures and low $|J|/t$ mentioned previously. To understand why, note that $l_{\text{ave}}$ diverges as the infinite temperature limit is approached, $\beta|J| \to 0$. However, the fluctuation length scale $l_{\text{fl}}$ remains finite as discussed in Sec. 4.1. Now, at sufficiently low $|J|/t$, $l_{\text{ave}}$ will eventually drop *below* the infinite temperature localization length going as $l_{\text{fl}} \propto (t/J)^2$, because it has a weaker $t/|J|$ scaling. Physically, the potential achieves a confining bias, $\langle V_\sigma \rangle (x) > 0$, which can localize the hole more strongly than the fluctutations at infinite temperatures. Finally, since the hole delocalizes asymptotically as zero temperature is reached in the ferromagnetic phase, the localization length is non-monotonic for low enough $|J|/t$.

In the ferromagnetic phase, I found in Fig. 2 that the fluctuation length scale $x_{\text{fl}}$ has the same scaling behavior with decreasing temperature as the average length scale $x_{\text{ave}}$. This means that $l_{\text{ave}}$ and $l_{\text{fl}}$ scale with temperature in the same manner, with exponential growth rates of $c_2 = 3$ and $c_2 = 5$ for the square and triangular lattices, respectively. This is seen to match the numerical findings in Fig. 4(c) in the regime of $|J| \ll t$.

Finally, we should understand why the scaling behavior is different for intermediate to large $|J|/t$. Here, it is important to again stress the difference with the two-leg ladder case [1]. There, as low temperatures are approached, one also approaches the phase transition from the para- to ferromagnetic phase. As a result, one can expect to only see a single length scale appear in this limit: the spin-spin correlation length. However, here as low temperatures are reached we do not approach a phase transition, because the system is already in the ferromagnetic phase. Therefore, there can easily be more than one length scale available. Indeed by analyzing the length scale over which a single spin flip occurs – see Appendix C – I find that this on average scales as

$$l_{\text{flip}} \propto \exp \left[ \frac{z}{2} \beta |J| \right], \tag{22}$$

for $z$ nearest neighbors. This gives *another* length scale with growth rates of $c_2 = 2$ and $c_2 = 3$ for the square and triangular lattices, respectively. At these length scales, the effective hole potential will, hereby, jump by $-|J|/2$ before jumping up again to 0. When $|J| \gg t$, this length scale will thus define the localization length, because the hole will reflect back, as soon as it meets this energetic barrier. This explains why $c_2 \to 2$ for the square lattice and $c_2 \to 3$ for the triangular lattice, when $|J|/t$ becomes large. As $|J|/t$ diminishes, however, the hole can start to tunnel through these barriers. Eventually, it can tunnel through enough barriers so that the lower length scale is given by either $l_{\text{ave}}$ or $l_{\text{fl}}$ in Eq. (21). This, consequently, explains the crossover between the two behaviors.

Finally, it is worth pointing out that at intermediate values $|J| \sim 3t$, the barriers are low enough that the hole can tunnel through many of them, but still the localization length is exponentially small compared to what one expects from Eq. (21). In this regime, it seems most accurate to think of the localization in terms of Anderson localization in the presence of weak disorder, i.e. when the disorder strength is smaller than or comparable to the kinetic energy. In such a scenario, instead of simple back-reflection, a particle localizes because it accumulates randomly varying phases for arriving to a particular point [23], i.e.

$$C(x) = c_1 e^{i\varphi_1} + c_2 e^{i\varphi_2} + \dots, \tag{23}$$

in which the phases $\varphi_1$ are basically chosen at random. In the present setup, these varying phases arise, because the distribution of the spin flips, happening on average on the length scale of $l_{\text{flip}}$, is random. Indeed, the standard deviation on $l_{\text{flip}}$ is on the same order as $l_{\text{flip}}$ itself, as shown in Appendix C. As a result, the hole will travel wildly different length scales between each barrier. As the hole can arrive between two barriers in many different ways, this gives rise to the destructive interference in Eq. (23).

The asymptotic delocalization of the hole in the low temperature limit describes a reversed metal-insulator *crossover*, in which the system is highly insulating at high temperatures, and becomes more and more metallic once the phase transition to the ferromagnetic phase is crossed and zero temperatures are approached.

## 4.3 Antiferromagnetic couplings

In this Section, we delve into the regime of antiferromagnetic couplings, $J > 0$. While ferromagnetic couplings led to qualitatively the same behavior for the square and triangular lattices, we shall see that antiferromagnetic couplings define highly distinct behaviors both for the underlying spin lattice and for the dopant dynamics.

The square lattice is bipartite, and may therefore be divided into sublattices A and B. This means that it is possible to rotate the local reference frame on every second site, such that $\hat{S}_j^{(z)} \to -\hat{S}_j^{(z)}$ on sublattice B. As the spin couplings are between nearest neighbors only, this also corresponds to flipping the sign of the spin coupling. This simple analysis shows that the ferromagnetic and antiferromagnetic cases are completely equivalent for a bipartite lattice. However, in the presence of holes and, as here, nearest neighbor hopping of the spins onto such vacant sites, the AFM and FM scenarios are no longer equivalent. Indeed, at low temperatures the staggered magnetization appearing in the Neél ordered ground state for AFM couplings gives rise to a *confined* hole as it starts to move. This naturally realizes an exact version of the retraceable path approximation due to Brinkman and Rice [7]. The confinement comes about, because the 1D motion of the hole realigns spins that were otherwise antialigned, giving rise to a linear potential increasing as $J/2 \cdot |x|$ [24]. The crossover from the thermally induced localization at high temperatures to the confined hole motion at low temperatures is illustrated in Fig. 5(left). Where the dynamics at high temperatures is mostly featureless, the motion at low temperatures is characterized by strong coherent oscillations. These have previously been shown to be due to interferences between the so-called string states that define

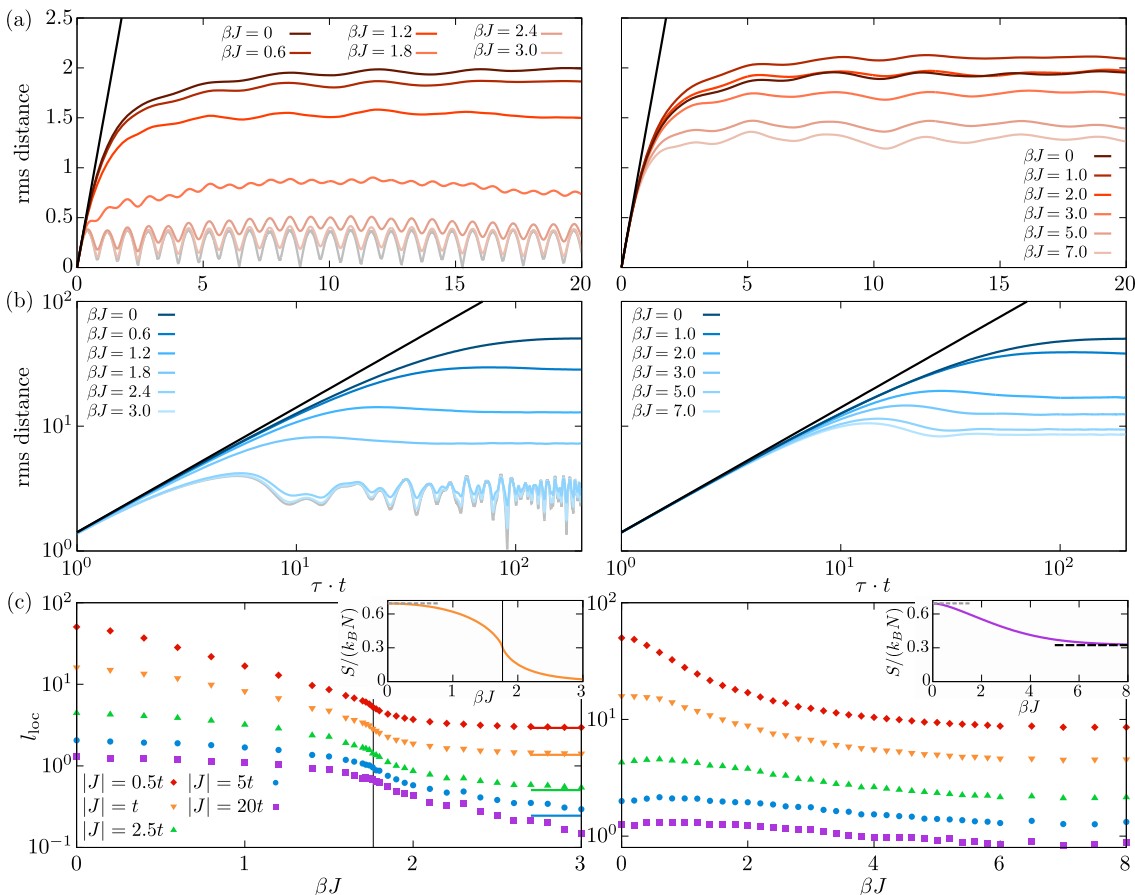

Figure 5: **Antiferromagnetic couplings.** Root-mean-square (rms) distance of hole to its original site as a function of time $\tau$ in units of hopping $t$ for $|J| = 5t$ (a) and $|J| = 0.5t$ (b) for indicated values of the inverse temperature $\beta$. For the square lattice (left), lower temperatures – higher $\beta|J|$ – results in more and more pronounced *coherent* oscillations eventually approaching the zero-temperature behavior in grey lines. For the triangular lattice (right), on the other hand, the dynamics depend only mildly on temperature, especially at larger spin couplings as in (a), and retains a thermal character even as zero temperature is approached. (c) Localization length versus inverse temperature for indicated values of the spin coupling for the square (left) and triangular (right) lattices. In the square lattice (right), the localization length decreases rapidly across the phase transition at $\beta_c^{\square}J = 2\ln(1 + \sqrt{2}) \simeq 1.76$ (vertical black line). In the triangular lattice (right), there is no phase transition and the localization length consequently has a much slower dependency on temperature. The insets in (c) show the entropy per particle $S/N$, both starting out at $S/N = k_B \ln(2)$ at high temperatures. The saturation at low temperatures happens as the entropy of the system approaches its zero-temperature limit (0 for the square lattice, $\simeq 0.323$ for the triangular lattice).

the low-energy eigenstates at zero temperature [24]. Moreover, in Fig. 5(c), we see that the approach to the zero-temperature limit happens as the entropy of the spin lattice [inset in Fig. 5(c)] approaches 0, around $\beta J \gtrsim 3$. The sharp decrease around the critical temperature behavior $\beta_c^{\square}J = 2\ln(1 + \sqrt{2})$ originates in this sense directly from the sharp decreasing behavior in the entropy at the phase transition.

The triangular lattice is, however, markedly different. In this case, the system is frustrated

and there is no mapping between the ferro- and antiferromagnetic cases. Even more dramatically, the frustration of the lattice leads to a non-vanishing entropy at zero temperature [25], shown in the inset of the right figure in Fig. 5(c). This strongly affects the dynamics as temperature is lowered. As mentioned in Sec. 4.1, the high-temperature limit gives the same dynamics both for ferro- and antiferromagnetic interactions *and* for the square and triangular geometries. However, as temperature is lowered there is, in stark contrast to the square lattice, no appearance of strong oscillations, as can be seen to the right in Figs. 5(a) and 5(b). The reason is that the ground state degeneracy is exponentially large in system size, meaning that the dynamics is averaged over many different spin realizations, even at the lowest temperatures, and this washes out the coherent oscillations that would otherwise appear. Moreover, the lack of a phase transition means that the localization length changes much slower with temperature as seen from the figure to the right in Fig. 5(c). In point of fact, we need to wait for the entropy per particle to be close to its zero-temperature limit, $S_0/N \simeq 0.323 k_B$, for the localization length of the hole to saturate to its zero-temperature limit. This only happens for inverse temperatures $\beta J \gtrsim 6$ for the triangular case.

What is also initially confounding about the triangular lattice is that there are ground state configurations, in which the effective hole potential is completely flat. Such configurations are all variations of the Neél states shown in Fig. 6(a). So if such configurations are there at low temperatures, why is the hole even localized? Surely, their presence must make it possible for the hole to escape its origin – even ballistically fast. Well, the answer to this conundrum turns out again to lie in the entropy of such states. As originally pointed out by Wannier [25], there are on the order of $2^{\sqrt{N}}$ states with the structure in Fig. 6(a). This scaling comes from realizing that the only alteration one can make to such a state is to shift the rows of the lattice by 1. And since there are $\sqrt{N}$ rows this gives $2^{\sqrt{N}}$ states. However, there is a much much larger family of states shown in Fig. 6(b). Here all the purple spins, which is every third, can be either spin-↑ or -↓ without a change in the energy. There are, therefore, at least a staggering $2^{N/3}$ of these[1]. This also explains why the entropy of the ground state manifold is nonzero. For just $N = 100$ spins, the relative abundance of the latter type of states to the former type is $2^{N/3}/2^{\sqrt{N}} \sim 10^7$, for 400 spins the ratio is at $\sim 10^{34}$! As a result, even though there are *in principle* states available in the ground state manifold in which the hole could delocalize, they have *zero* statistical weight.

## 4.4 Spin coupling scaling dependency

Before concluding, this Section is concerned with describing in detail the dependency of the localization length on the ratio between the spin coupling and the hopping amplitude, $J/t$. Examples of this dependency are shown in Figs. 7(a) and 7(b) for ferro- and antiferromagnetic couplings, respectively. This is furthermore compared to the universal behavior found at infinite temperatures, identical for the square and triangular lattices and for both ferro- and antiferromagnetic spin couplings, scaling as $(t/J)^2$ for $|J|/t \ll 1$. Analogous to the two-leg ladder [1], this scaling behavior turns out to be highly specific to the infinite-temperature limit. In point of fact, for any finite temperature we observe that the asymptotic behavior is rather $t/|J|$ with a temperature dependent prefactor. This is because as $t/|J|$ is lowered, the bias of the effective hole potential will eventually dominate over the fluctuations, i.e. $l_{\mathrm{ave}} \sim t/|J| < l_{\mathrm{fl}} \sim (t/J)^2$ for sufficiently small $|J|/t$, no matter the temperature.

Moreover, we see that for ferromagnetic couplings in Fig. 7(a), equal values of $\beta/\beta_c^{\square}$ and $\beta/\beta_c^{\triangle}$ – i.e. temperatures in the same proportion to their respective critical temperatures – are quantitatively very similar, especially for low values of $|J|/t$. On the antiferromagnetic side in

---

[1] Wannier [25] came up with an even stronger lower bound of $2^{5N/12}$ based on simple geometrical arguments. This gives a ground state entropy $> 5\ln(2)/12 k_B N \simeq 0.289 k_B N$, pretty close to the exact value of $S_0 \simeq 0.323 k_B N$.

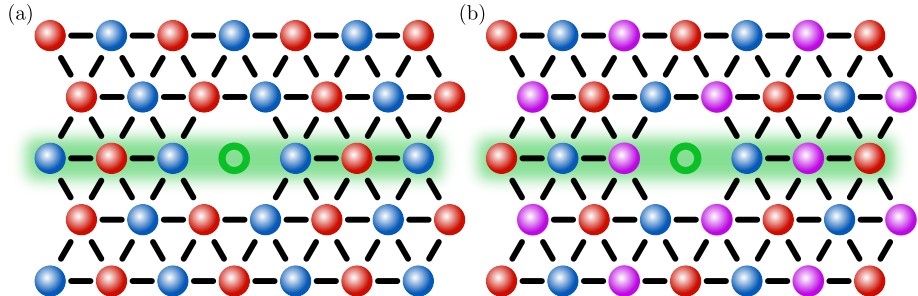

Figure 6: Origin of localization for triangular lattice at low temperatures. (a) A perfect Neél structure is a part of the ground state manifold. As the hole moves through such a state, it experiences a completely flat potential, $V(x) = 1/2$ for $x \neq 0$, since the perpendicular part vanishes $V_\perp(x) = 0$. This suggests that the hole should be able to delocalize for antiferromagnetic couplings. However, the number of perfect Neél ordered states scales only as $2^{\sqrt{N}}$, and there are configurations (b) that have a *much* larger weight. Here, all purple spins can be chosen freely between $|\uparrow\rangle, |\downarrow\rangle$, leading to at least $2^{N/3}$ states. For these, the hole always experiences an overall growing potential, as it is guaranteed to increase for every third hop. As the latter type in (b) completely outnumbers the first type in (a), the hole remains localized.

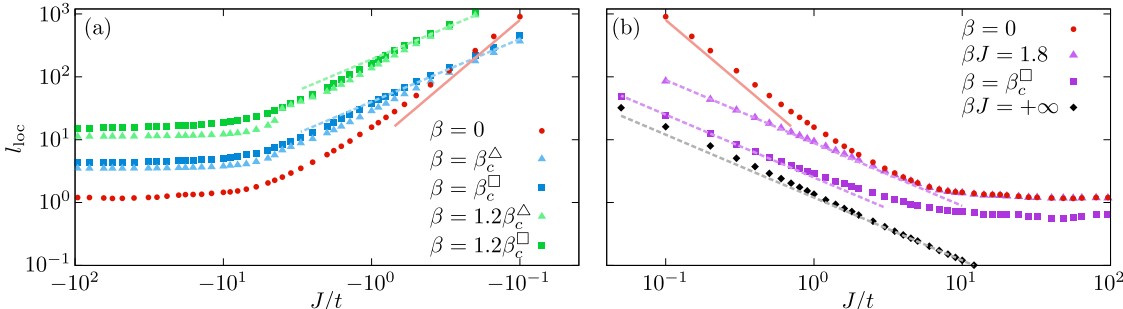

Figure 7: Localization length dependency for ferromagnetic (a) and antiferromagnetic (b) couplings as a function of $J/t$ on a log-log plot for indicated inverse temperatures. Square and triangular symbols are for the square and triangular lattices, respectively. Also, $\beta_c^\square = 2\ln(1+\sqrt{2})/|J|$ and $\beta_c^\triangle = \ln(3)/|J|$ indicate the inverse critical temperatures in the square and triangular lattice, respectively. Only at infinite temperature is the asymptotic scaling $(t/J)^2$ (solid red line). For any finite temperature, the asymptotic behavior (dashed lines) is $t/|J|$ with a temperature dependent prefactor.

Fig. 7(b), we additionally observe that for the square lattice, the localization length already at the critical temperature is quantitatively close to the zero-temperature limit for low $J/t$, with a factor of 1.5 between them. For larger $J/t$, the localization length is short, and it becomes important that the system locally has occasional spin flips with respect to the perfect Neél-ordered state at zero temperature. Finally, for hole motion in the triangular lattice the effective hole potential in the ground state manifold only increases in every third hop, as described in Fig. 6. As a result, the localization length is strictly larger for the hole in the triangular lattice compared to the square lattice at similar temperatures, here shown for $\beta J = 1.8 \simeq \beta_c^\square J$. Also, at these temperatures the localization length of the hole in the triangular lattice even follows the infinite temperature behavior for $J \gtrsim 2t$.

# 5   Discussion

In this Section, I will discuss generalizations to the mixed-dimensional $t$-$J_z$ model considered in this Article.

## 5.1   Beyond 1D dopant motion

A central assumption in the present Article has been the one-dimensional nature of the dopant motion. This not only enables the numerical computation of the results to large system sizes and long times, but also clarifies certain physical situations that are far more complex in higher dimensions. For example, when the hole moves solely in 1D it has to scatter on any domain walls it may find on its way. If the hole were to move in 2D, it could very well be able to circumvent these, enabling a slow, but continuing, propagation. In the ferromagnetic phase, where the system macroscopically occupies one spin state, say spin-$|\uparrow\rangle$, the domains of spin-$|\downarrow\rangle$ become smaller and rarer, and eventually consist of singly flipped spins as discussed in Sec. 4.2. In this low-temperature ferromagnetic limit, the system hereby resembles a perfect lattice with occasional "impurities" of spin-$|\downarrow\rangle$ that the hole may scatter on. This seems essentially equivalent to the propagation of electrons in lattices with a low density of defects – or local impurities. As a result, in the ferromagnetic phase we should at the very least expect the hole to perform diffusion with *weak localization* corrections [36], if not full-scale ballistic motion.

Contrary to the usual case in the solid state, however, the effective defect density – the domain walls – in the present setup increases rapidly with temperature. As a result, the mean free path decreases to be on on the order of the lattice spacing at high temperatures. In this limit, therefore, scattering happens all the time and the dopant may very well localize completely. These considerations illustrate that the inverted metal-insulator *crossover* found in the present analysis for 1D dopant motion may be replaced by an actual *transition* for 2D motion, and should probably happen at the underlying Curie temperature of the spin lattice.

Detailed numerical analyses in the infinite temperature limit of vanishing spin couplings for 2D motion of dopants [2, 3] illustrate that the situation even in this restricted setup is quite complex. At the short to intermediate timescales investigated, the motion is a lot slower than ballistic motion, but remain faster than the predictions of the retraceable path approximation [2, 7], and faster than expected from the mapping to a Bethe lattice [3]. Conclusions on long timescales, however, remain to be drawn and must await further analyses.

## 5.2   Beyond the Ising model

Another crucial assumption in the presented analysis is the Ising type spin couplings. One could go beyond this realm by introducing spin flip-flop terms into the Hamiltonian. The computational complexity in this case, however, increases in a daunting manner, as the description even just of the underlying spin lattice now becomes highly complex and – at best – approximate. At low temperatures and antiferromagnetic couplings, there is good evidence that quasiparticles – magnetic polarons – form [9]. Indeed, linear spin-wave theory combined with the selfconsistent Born approximation has been shown to compare well to exact diagonalization studies [10, 37] and Monte-Carlo simulations [38]. Even more importantly, it has successfully explained [35] the experimentally observed propagation of holes [16] through the formation and propagation of such magnetic polarons. As temperature is increased, however, there are currently only limited approaches available [39], and the quasiparticle picture at low temperatures is even still debated [40–45].

One possible path forward could be to go in the opposite regime of the XY model, where the *only* present spin couplings are flip-flop terms. In one dimension, this supports a simple analytical solution via the Jordan-Wigner transformation [46]. If one can also formulate the

motion of dopants in an efficient manner in this model, this would certainly be an interesting pathway to pursue.

### 5.3   Increasing the doping level

In the present analysis, I have focused on the propagation of a single dopant. One may wonder, what happens as the doping level rises. If their initial mutual distances are much larger than the single dopant localization length, one can expect them to retain their single-dopant characteristics uncovered in the present analysis. However, if two dopants start out close to each another, they may significantly alter each others motion. This could lead to novel phenomena, as their motion may become strongly correlated even though they are submerged in, e.g., an infinite temperature spin environment.

### 5.4   Connecting to an external bath

A final obvious extension of the ideas pursued in the present Article is to investigate the same type of dynamics, but in the presence of a coupling to an external heat bath. This should drive the system back to thermal equilibrium by allowing spin flip dynamics to occur, and may be a way to investigate their influence in a simpler manner than through spin flip-flop terms in the Hamiltonian. It is clear, however, that the methodology must be changed substantially in such an open system case, as now the propagation dynamics cannot be calculated as the thermal – Boltzmann-weighted – average of pure state evolutions.

## 6   Conclusions and outlook

In this Article, I have investigated the one-dimensional motion of a dopant in two-dimensional square and triangular lattices of Ising coupled spins. The thermally induced localization effect originally found in a two-leg ladder geometry [1] has, hereby, been extended to the case in which there exists a finite temperature transition to a long-range ordered ferromagnetic phase. While the high-temperature limit features universal localized hole dynamics across ferro- and antiferromagnetic couplings as well as the two investigated lattice geometries, finite temperatures break this correspondence. On the ferromagnetic side, the hole remains localized across the Curie temperature and feature very similar behaviors for the two investigated geometries. While the localization in the two-leg ladder was found to scale identically with temperature across any (negative) value of $J/t$, this is no longer true for the two-dimensional system. At small $|J|/t$, the localization can be understood as the back-scattering off an effective hole potential that fluctuates to large values as the hole moves away from its origin. At large $|J|/t$, the hole instead back-reflects on singular spin flips happening on an exponentially shorter length scale. Inbetween, increased tunneling through these singular spin flips describes a crossover from one to the other behavior as $|J|/t$ is lowered.

On the antiferromagnetic side, only the square lattice features a phase transition to a long-range ordered AFM phase. As the antiferromagnetic correlations grow, the hole experiences a stronger and stronger linear potential, because its motion now starts to realign spins that were otherwise antialigned. This leads to a crossover between (thermal) disorder induced localization to confinement, with a particularly sharp decrease of the localization length around the Curie temperature in the square lattice. The characteristics of the dynamics in these two regimes is also markedly different. At high temperatures, the motion is incoherent with a smooth, featureless behavior of the root-mean-square distance of the hole to its origin. As zero temperature is approached, and the entropy of the system vanishes, stronger coherent

oscillation occur due to quantum interference of the low-lying energy states, the so-called string states [24, 47].

In the triangular lattice, no phase transition happens for antiferromagnetic couplings due to frustration. This makes the approach to the zero-temperature limit much slower, explained well by when the entropy drops to its nonzero zero-temperature limit [25]. The associated exponentially large ground state manifold means that the hole dynamics retains its featureless, thermal behavior.

These detailed investigations show that the motion of dopants, even in these highly simplistic models, host rich and diverse behaviors, in which an indepth knowledge of the underlying spin lattice is crucial for understanding the dopant dynamics. For ferromagnetc couplings, it describes an intriguing *reversed* metal-insulator crossover, from a highly insulating regime at high temperatures to an increasingly metallic regime at low temperatures. Moreover, there are several exciting research pathways that may be undertaken to expand these considerations. First and foremost, it would be interesting to study the stability of the localization effect. Here, one could study the influence of coupling the spins to an external heat bath driving them towards thermalization. In such a scenario, the thermal fluctuations might play a similar role to flip-flop spin interactions, and such links could be pursued further. One could also introduce such flip-flop terms in the Hamiltonian directly, and finally one could pursue the understanding of less restrained hole motion, where it is allowed to move not only along a one-dimensional line in the lattice.

# Acknowledgements

The author thanks J. Ignacio Cirac, Pavel Kos, Dominik S. Wild, and Marton Kanasz-Nagy for valuable discussions.

**Funding information**  This article was supported by the Carlsberg Foundation through a Carlsberg Internationalisation Fellowship, grant number CF21_0410.

# A Short-range correlators: square lattice

In this Appendix, I compute the nearest and next-nearest spin correlators for the square lattice. The calculation is based Ref. [48]. Starting from the Hamiltonian

$$\hat{H}_J = -|J| \sum_{\langle \mathbf{i},\mathbf{j} \rangle} \hat{S}_{\mathbf{i}}^{(z)} \hat{S}_{\mathbf{j}}^{(z)} = -\frac{|J|}{4} \sum_{\langle \mathbf{i},\mathbf{j} \rangle} \hat{s}_{\mathbf{i}} \hat{s}_{\mathbf{j}}, \tag{A.1}$$

we can express the desired correlators as

$$C(1) = 4 \langle \hat{S}_{0,0}^{(z)} \hat{S}_{1,0}^{(z)} \rangle = \langle \hat{s}_{0,0} \hat{s}_{1,0} \rangle, \quad C(\sqrt{2}) = 4 \langle \hat{S}_{0,0}^{(z)} \hat{S}_{1,1}^{(z)} \rangle = \langle \hat{s}_{0,0} \hat{s}_{1,1} \rangle. \tag{A.2}$$

Here, I define $\hat{s} = 2\hat{S}^{(z)}$, such that it can take on the values $\pm 1$. The nearest and next-nearest neighbor correlators are computed from the expression

$$a_0(\alpha_1, \alpha_2) = \int_0^{2\pi} \frac{d\theta}{2\pi} \left[ \frac{(1 - \alpha_1 e^{i\theta})(1 - \alpha_2 e^{-i\theta})}{(1 - \alpha_1 e^{-i\theta})(1 - \alpha_2 e^{+i\theta})} \right]^{1/2}. \tag{A.3}$$

Here, the only difference between the two correlators are in the choice of the $\alpha_i$. Explicitly,

$$C(1) : \alpha_1 = e^{-\beta|J|/2} \tanh\left(\frac{\beta|J|}{4}\right), \ \alpha_2 = e^{-\beta|J|/2} \coth\left(\frac{\beta|J|}{4}\right),$$

$$C(\sqrt{2}) : \alpha_1 = 0, \ \alpha_2 = \frac{1}{\sinh^2\left(\frac{\beta|J|}{2}\right)}. \tag{A.4}$$

This allows me to numerically compute these correlators. Also, a rather tedious low-temperature expansion shows that

$$C(1) \to 1 - 4e^{-2\beta|J|} - \frac{47}{4} e^{-3\beta|J|},$$

$$C(\sqrt{2}) \to 1 - 4e^{-2\beta|J|} - 16e^{-3\beta|J|}. \tag{A.5}$$

The length scale arising from the mean value of the hole potential then asymptotically scales as

$$x_{\text{ave}} = \frac{2}{C(1) - C(\sqrt{2})} \to \frac{8}{64 - 47} e^{+3\beta|J|} = \frac{8}{17} e^{+3\beta|J|}, \tag{A.6}$$

having a fast $e^{+3\beta|J|}$ scaling behavior.

# B Short-range correlators: triangular lattice

In this Appendix, I compute the nearest and next-nearest neighbor spin correlators for ferromagnetic couplings in the triangular lattice. The calculation is based on Refs. [49, 50]. The setup is identical to the one in the previous Appendix, albeit with the diagonal coupling appropriate for the triangular lattice. I also define $\nu = \tanh(\beta|J|/4)$. I need the nearest and next-nearest neighbor correlators (at distance 1 and $\sqrt{13/4}$)

$$C(1) = 4 \langle \hat{S}_{0,0}^{(z)} \hat{S}_{1,0}^{(z)} \rangle = \langle \hat{s}_{0,0} \hat{s}_{1,0} \rangle, \quad C(\sqrt{13/4}) = 4 \langle \hat{S}_{1,0}^{(z)} \hat{S}_{0,1}^{(z)} \rangle = \langle \hat{s}_{1,0} \hat{s}_{0,1} \rangle. \tag{B.1}$$

The nearest-neighbor correlator is explicitly computed in Ref. [49] to be

$$C(1) = \int_{-\pi}^{\pi} \frac{d\omega}{2\pi} \left[ \frac{a - be^{+i\omega} - ce^{-i\omega}}{a - be^{-i\omega} - ce^{+i\omega}} \right]^{1/2}, \tag{B.2}$$

with

$$a = 2v(1 + v^2), \; b = v^2 c = v^2(1 - v)^2. \tag{B.3}$$

The next-nearest correlator $C(\sqrt{13/4})$ does not seem to be explicitly computed in Stephenson's papers. However, we may use the results for 4-point correlators to get $C(\sqrt{13/4})$. In particular, slightly rewritting Eq. (2.15) in Ref. [50], I get

$$\langle \hat{s}_{0,0} \hat{s}_{1,0} \hat{s}_{p,q} \hat{s}_{p,q+1} \rangle = C(1)^2 + (1 + v)^2 (1 - v)^2$$
$$\times \left\{ [p - 1, q]_{4,3} [p, q + 1]_{1,6} - [p, q]_{1,3} [p - 1, q + 1]_{4,6} \right\}. \tag{B.4}$$

Here, the notation $[p, q]$ is short-hand for the $6 \times 6$ matrix

$$[p, q] = A^{-1}(p, q) = \int_{-\pi}^{\pi} \frac{d\varphi_1}{2\pi} \int_{-\pi}^{\pi} \frac{d\varphi_2}{2\pi} e^{-i(p\varphi_1 + q\varphi_2)} A^{-1}(\phi_1, \phi_2). \tag{B.5}$$

Here, $A(\phi_1, \phi_2)$ is a specific $6 \times 6$ matrix depending on $v, \varphi_1, \varphi_2$, which I will return to in a moment. Inserting $p = q = 0$ in Eq. (B.4), the 4-point correlator collapses to $\langle \hat{s}_{0,0} \hat{s}_{1,0} \hat{s}_{0,0} \hat{s}_{0,1} \rangle = \langle \hat{s}_{1,0} \hat{s}_{0,1} \rangle = C(\sqrt{2})$, as the $\hat{s}$ operators commute and $\hat{s}_{p,q}^2 = 1$. In this manner,

$$C(\sqrt{13/4}) = C(1)^2 + (1 + v)^2 (1 - v)^2 \left\{ [-1, 0]_{4,3} [0, +1]_{1,6} - [0, 0]_{1,3} [-1, +1]_{4,6} \right\}. \tag{B.6}$$

The matrix that we need to invert is given in Eq. (2.4) in Ref. [50]

$$A(\varphi_1, \varphi_2) = \begin{bmatrix} 0 & 1 & 1 & 1 - v e^{i\varphi_1} & 1 & 1 \\ -1 & 0 & 1 & 1 & 1 - v e^{i(\varphi_1 + \varphi_2)} & 1 \\ -1 & -1 & 0 & 1 & 1 & 1 - v e^{i\varphi_2} \\ -1 + v e^{-i\varphi_1} & -1 & -1 & 0 & 1 & 1 \\ -1 & -1 + v e^{-i(\varphi_1 + \varphi_2)} & -1 & -1 & 0 & 1 \\ -1 & -1 & -1 + v e^{-i\varphi_2} & -1 & -1 & 0 \end{bmatrix}. \tag{B.7}$$

Here, I perform the inversion of the matrix in Mathematica. I express the result in terms of the cofactor matrix $C$: $A^{-1} = C^T / \Delta$, where $C^T$ is the transpose of $C$ and $\Delta = \det(A)$ is the determinant. Explicitly,

$$\Delta = (1 + v)^2 \left[ \frac{(1 + v^2)^3 + 8v^3}{(1 + v)^2} - 2v(1 - v)^2 \left\{ \cos(\varphi_1) + \cos(\varphi_2) + \cos(\varphi_1 + \varphi_2) \right\} \right] \tag{B.8}$$

Transforming the variables as $\theta = -\varphi_2, \omega = \varphi_1 + \varphi_2$, one can express the determinant in the form

$$\Delta(\theta, \omega) = (1 + v)^2 \left[ A + B \cos(\theta) + C \sin(\theta) \right]. \tag{B.9}$$

Here,

$$A = \frac{(1 + v^2)^3 + 8v^3}{(1 + v)^2} - 2v(1 - v)^2 \cos(\omega), \tag{B.10}$$

$$B = -2v(1 - v)^2 \left[ 1 + \cos(\omega) \right], \tag{B.11}$$

$$C = 2v(1 - v)^2 \sin(\omega). \tag{B.12}$$

Moreover, we need combonents $C_{3,4}, C_{6,1}, C_{3,1}$ and $C_{6,4}$ to compute the correlator. First,

$$C_{3,4}(\varphi_1, \varphi_2) = (1 + v) \left\{ [1 - v(1 - 2v)] - v[2 - v(1 - v)] e^{-i(\varphi_1 + \varphi_2)} - v(1 - v) \left[ e^{-i\varphi_1} + e^{-i\varphi_2} \right] \right\}$$
$$= (1 + v) \left\{ [1 - v(1 - 2v)] - v[2 - v(1 - v)] e^{-i\omega} - v(1 - v) \left[ e^{-i(\omega + \theta)} + e^{i\theta} \right] \right\}$$
$$= (1 + v) \left\{ D + E e^{-i\omega} + F \left[ e^{-i(\omega + \theta)} + e^{i\theta} \right] \right\}, \tag{B.13}$$

with

$$D = [1 - v(1 - 2v)], \; E = -v[2 - v(1 - v)], \; F = -v(1 - v). \tag{B.14}$$

Second,

$$C_{6,1}(\varphi_1, \varphi_2) = -(1 + v)\left\{ D + E e^{+i(\varphi_1 + \varphi_2)} + F\left[ e^{+i\varphi_1} + e^{+i\varphi_2} \right] \right\} = -C_{3,4}^*(\varphi_1, \varphi_2)$$

$$= -(1 + v)\left\{ D + E e^{+i\omega} + F\left[ e^{+i(\omega + \theta)} + e^{-i\theta} \right] \right\}. \tag{B.15}$$

Third,

$$C_{3,1}(\varphi_1, \varphi_2) = -(1 + v)\left\{ -(1 - v) + v^2(1 - v)e^{+i(\varphi_1 - \varphi_2)} + v(1 + v)\left[ e^{+i\varphi_1} + e^{-i\varphi_2} \right] \right\}$$

$$= -(1 + v)\left\{ -(1 - v) + v^2(1 - v)e^{+i(\omega + 2\theta)} + v(1 + v)e^{i\theta}\left[ e^{+i\omega} + 1 \right] \right\}. \tag{B.16}$$

And finally,

$$C_{6,4}(\varphi_1, \varphi_2) = -(1 + v)\left\{ -(1 - v) + v^2(1 - v)e^{-i(\varphi_1 - \varphi_2)} + v(1 + v)\left[ e^{-i\varphi_1} + e^{+i\varphi_2} \right] \right\} = C_{3,1}^*(\varphi_1, \varphi_2)$$

$$= -(1 + v)\left\{ -(1 - v) + v^2(1 - v)e^{-i(\omega + 2\theta)} + v(1 + v)e^{-i\theta}\left[ e^{-i\omega} + 1 \right] \right\}. \tag{B.17}$$

Now, I will the terms in Eq. (B.6) explicitly. First,

$$[-1, 0]_{4,3} = \int_{-\pi}^{\pi} \frac{d\varphi_1}{2\pi} \int_{-\pi}^{\pi} \frac{d\varphi_2}{2\pi} e^{i\varphi_1} \frac{C_{3,4}(\varphi_1, \varphi_2)}{\Delta(\varphi_1, \varphi_2)} = \int_{-\pi}^{\pi} \frac{d\omega}{2\pi} \int_{-\pi}^{\pi} \frac{d\theta}{2\pi} e^{i(\omega + \theta)} \frac{C_{3,4}(\omega, \theta)}{\Delta(\omega, \theta)}$$

$$= \frac{1}{1 + v} \int_{-\pi}^{\pi} \frac{d\omega}{2\pi} \int_{-\pi}^{\pi} \frac{d\theta}{2\pi} \frac{F + (De^{i\omega} + E)e^{i\theta} + Fe^{i\omega}e^{2i\theta}}{A + B\cos(\theta) + C\sin(\theta)}$$

$$= \frac{1}{1 + v} \int_{-\pi}^{\pi} \frac{d\omega}{2\pi} \left[ F I_0(\omega) + (De^{i\omega} + E)I_1(\omega) + Fe^{i\omega}I_2(\omega) \right]. \tag{B.18}$$

Here, I follow Stephenson [49] and define

$$I_n(\omega) = \int_{-\pi}^{\pi} \frac{d\theta}{2\pi} \frac{e^{in\theta}}{A + B\cos(\theta) + C\sin(\theta)} = \frac{\alpha^n}{(A^2 - B^2 - C^2)^{1/2}}. \tag{B.19}$$

Importantly, this integral is solved exactly. Here, $\alpha = [(A^2 - B^2 - C^2)^{1/2} - A]/[B - iC]$. I, furthermore, define $I_{n,m} = \int_{-\pi}^{\pi} d\omega e^{im\omega} I_n(\omega)/(2\pi)$. Then

$$[-1, 0]_{4,3} = \frac{1}{1 + v} \int_{-\pi}^{\pi} \frac{d\omega}{2\pi} \left[ F I_0(\omega) + (De^{i\omega} + E)I_1(\omega) + Fe^{i\omega}I_2(\omega) \right]$$

$$= \frac{1}{1 + v} \left[ D I_{1,1} + E I_{1,0} + F(I_{0,0} + I_{2,1}) \right]. \tag{B.20}$$

Likewise,

$$[0, 1]_{1,6} = \int_{-\pi}^{\pi} \frac{d\varphi_1}{2\pi} \int_{-\pi}^{\pi} \frac{d\varphi_2}{2\pi} e^{-i\varphi_2} \frac{C_{6,1}(\varphi_1, \varphi_2)}{\Delta(\varphi_1, \varphi_2)} = \int_{-\pi}^{\pi} \frac{d\omega}{2\pi} \int_{-\pi}^{\pi} \frac{d\theta}{2\pi} e^{i\theta} \frac{C_{6,1}(\omega, \theta)}{\Delta(\omega, \theta)}$$

$$= -\frac{1}{1 + v} \int_{-\pi}^{\pi} \frac{d\omega}{2\pi} \int_{-\pi}^{\pi} \frac{d\theta}{2\pi} \frac{F + (D + Ee^{i\omega})e^{i\theta} + Fe^{i\omega}e^{2i\theta}}{A + B\cos(\theta) + C\sin(\theta)}$$

$$= -\frac{1}{1 + v} \int_{-\pi}^{\pi} \frac{d\omega}{2\pi} \left[ F I_0(\omega) + (D + Ee^{i\omega})I_1(\omega) + Fe^{i\omega}I_2(\omega) \right]$$

$$= -\frac{1}{1 + v} \left[ D I_{1,0} + E I_{1,1} + F(I_{0,0} + I_{2,1}) \right] \tag{B.21}$$

which is very similar to $[-1, 0]_{4,3}$. Moreover,

$$
\begin{aligned}
[0, 0]_{1,3} &= \int_{-\pi}^{\pi} \frac{d\omega}{2\pi} \int_{-\pi}^{\pi} \frac{d\theta}{2\pi} \frac{C_{3,1}(\omega, \theta)}{\Delta(\omega, \theta)} \\
&= -\frac{1}{1+v} \left[ -(1-v)I_{0,0} + v^2(1-v)I_{2,1} + v(1+v)(I_{1,1} + I_{1,0}) \right].
\end{aligned}
\tag{B.22}
$$

And finally,

$$
\begin{aligned}
[-1, +1]_{4,6} &= \int_{-\pi}^{\pi} \frac{d\omega}{2\pi} \int_{-\pi}^{\pi} \frac{d\theta}{2\pi} e^{i(\omega + 2\theta)} \frac{C_{6,4}(\omega, \theta)}{\Delta(\omega, \theta)} \\
&= -\frac{1}{1+v} \left[ -(1-v)I_{2,1} + v^2(1-v)I_{0,0} + v(1+v)(I_{1,1} + I_{1,0}) \right].
\end{aligned}
\tag{B.23}
$$

So, to compute $C(\sqrt{13/4})$, I need to compute the four terms $I_{0,0}, I_{1,0}, I_{1,1}, I_{2,1}$.

## C   Localization length in the ferromagnetic phase for large $|J|/t$

In this Appendix, I derive the average distance between spin flips in the ferromagnetic phase. I then use this to calculate what the asymptotic localization length is at low temperatures and large $|J|/t$.

First, in a lattice with $z$ nearest neighbor interactions, the probability to have a single spin flip is proportional to the Boltzmann factor $p_{\text{flip}} = e^{-z\beta|J|/2}$. Here $z = 4, 6$ correspond to the square and triangular lattices, respectively. This means that at low temperatures, the probability to find a strip of length $l \geq 1$ with exactly one spin flip at the end is proportional to $p_{\text{flip}}[1 - p_{\text{flip}}]^{l-1}$. Since the normalization constant is simply unity, $A = \sum_{l=1}^{\infty} p_{\text{flip}}[1 - p_{\text{flip}}]^{l-1} = p_{\text{flip}} \sum_{l=0}^{\infty}[1 - p_{\text{flip}}]^l = 1$, the probability to find such a segment of length $l$ is

$$
P(l) = p_{\text{flip}}[1 - p_{\text{flip}}]^{l-1}.
\tag{C.1}
$$

The average length of such a segment gives the mean distance between spin flips (along a line) at low temperatures

$$
\begin{aligned}
l_{\text{flip}} = \langle l \rangle &= \sum_{l=1}^{\infty} l P(l) = p_{\text{flip}} \sum_{l=1}^{\infty} l[1 - p_{\text{flip}}]^{l-1} = p_{\text{flip}} \frac{d}{dr} \sum_{l=0}^{\infty} r^l \bigg|_{r=1-p_{\text{flip}}} = p_{\text{flip}} \frac{d}{dr} \frac{1}{1-r} \bigg|_{r=1-p_{\text{flip}}} \\
&= p_{\text{flip}} \frac{1}{p_{\text{flip}}^2} = p_{\text{flip}}^{-1} = e^{z\beta|J|/2}.
\end{aligned}
\tag{C.2}
$$

Now, spin flips along any of the three lines $y = -1, 0, +1$ will give a change of $-|J|/2$ in the potential. However, at low temperatures we may treat the legs as independent. As a result, taking the other legs into account will not alter this asymptotic result.

Second, I now use this average distance to calculate the localization length for $|J|/t \gg 1$. In this limit, the hole is effectively a single particle in a one-dimensional infinite square well potential of length $l_{\text{flip}}$. Now, the initial state is centered at $x = 0$. Since it is on a single lattice site, in this continuum limit I will take it to be a constant with width 1 (in units of the lattice spacing)

$$
\Psi(x, t = 0) = 1, \quad -1/2 \leq x \leq +1/2.
\tag{C.3}
$$

Since this is even, there is only an overlap with the even eigenfunctions, $\psi_{2n+1}(x) = \sqrt{2/l_{\text{flip}}}$ $\cos[(2n+1)\pi x/l_{\text{flip}}]$, where $n = 0, 1, 2, \ldots$. The overlaps are then

$$c_{2n+1} = \langle \psi_{2n+1}|\Psi(t=0)\rangle = \int_{-1/2}^{+1/2} dx\, \psi_{2n+1}(x) = \frac{2\sqrt{2l_{\text{flip}}}}{(2n+1)\pi} \sin\left[\frac{(2n+1)\pi}{2l_{\text{flip}}}\right]. \qquad \text{(C.4)}$$

To compute the asymptotic mean-square distance, $\langle x^2 \rangle = \sum_n |c_{2n+1}|^2 \langle \psi_{2n+1}|x^2|\psi_{2n+1}\rangle$, we need the mean-square distance for each of the contributing eigenfunctions. These are

$$\langle \psi_{2n+1}|x^2|\psi_{2n+1}\rangle = \frac{2}{l_{\text{flip}}} \int_{-l_{\text{flip}}/2}^{+l_{\text{flip}}/2} dx\, x^2 |\psi_{2n+1}(x)|^2 = \frac{l_{\text{flip}}^2}{4}\left[\frac{1}{3} - \frac{2}{(2n+1)^2\pi^2}\right]. \qquad \text{(C.5)}$$

The resulting asymptotic mean-square distance is

$$\langle x^2 \rangle = \sum_n |c_{2n+1}|^2 \langle \psi_{2n+1}|x^2|\psi_{2n+1}\rangle = \sum_n |c_{2n+1}|^2 \frac{l_{\text{flip}}^2}{4}\left[\frac{1}{3} - \frac{2}{(2n+1)^2\pi^2}\right] \to \frac{l_{\text{flip}}^2}{12}. \qquad \text{(C.6)}$$

Here, I use that the $1/(2n+1)^2$ term will contribute with a term linear in $l_{\text{flip}}$, which is exponentially small compared to $l_{\text{flip}}$ at low temperatures. Note that this asymptotic behavior is actually independent of the particular choice of the initial state in Eq. (C.3), because only the common $l_{\text{flip}}^2/12$ part for the mean-square distance is retained. Hence, the asymptotic rms distance is

$$x_{\text{rms}} \to \sqrt{\langle x^2 \rangle} = \frac{l_{\text{flip}}}{2\sqrt{3}} = \frac{e^{z\beta|J|/2}}{2\sqrt{3}}. \qquad \text{(C.7)}$$

This is valid in the limit of strong spin couplings, $|J|/t \gg 1$, and at low temperatures, $\beta|J| \gg 1$.

# D   Entropy for antiferromagnetic couplings

In this Appendix, I calculate the entropy as a function of temperature for antiferromagnetic couplings, both for the square and triangular lattice.

The calculation is carried through by using the thermodynamic relation $F = U - TS$, between the free energy $F$, the average energy $U = \langle \hat{H}_J \rangle$, and the entropy, $S$. In particular, both for the square and triangular lattice there are explicit expressions for $F$ and $U$, whereby the entropy per particle may readily be computed as

$$\frac{S}{k_B N} = \frac{\beta}{N}[U - F], \qquad \text{(D.1)}$$

with $\beta = 1/(k_B T)$ the inverse temperature. For the square lattice in particular [51, 52],

$$-\beta\frac{F}{N} = \ln\left[2\cosh\left(\frac{\beta|J|}{2}\right)\right] + \frac{1}{\pi}\int_0^{\pi/2} d\varphi \ln[f(k,\varphi)], \qquad \text{(D.2)}$$

with $f(k,\varphi) = \frac{1}{2}\left\{1 + (1 - k^2(\beta|J|)\sin^2\varphi)^{1/2}\right\}$ and $k(\beta|J|) = 2\sinh(\beta|J|/2)/\cosh^2(\beta|J|/2)$. From the relation $U = \langle \hat{H}_J \rangle = -\partial_\beta \ln(Z)$, where $Z = \text{tr}[e^{-\beta\hat{H}_J}] = e^{-\beta F}$ is the partition function, it follows that $F = -\ln(Z)/\beta$, and hereby

$$\frac{U}{N|J|} = -\partial_{\beta|J|}\left[-\frac{\beta F}{N}\right] =$$

$$\frac{1}{2}\tanh[\frac{\beta|J|}{2}] + \frac{1 - 2\tanh(\beta|J|/2)}{\cosh(\beta|J|/2)}k(\beta|J|)\frac{1}{\pi}\int_0^{\pi/2} d\varphi f^{-1}(k,\varphi)(1 - k^2\sin^2\varphi)^{-1/2}. \qquad \text{(D.3)}$$

The integrals appearing in Eqs. (D.2) and (D.3) are easily solved numerically. By insertion in Eq. (D.1), we hereby have the entropy per particle for the square Ising lattice.

For the triangular lattice, there are also exact expressions. The free energy can be calculated from [25]

$$-\beta \frac{F}{N} = \ln\left[ 2e^{-\beta J/4} \cosh\left(\frac{\beta J}{2}\right)\right]$$
$$+ \frac{1}{2\pi^2}\int_0^\pi d\omega_1 \int_0^\pi d\omega_2 \ln\left[ 1 + 4\kappa \cos(\omega_1)\cos(\omega_2) - 4\kappa \cos^2(\omega_2)\right], \qquad (D.4)$$

where $\kappa(\beta J) = [e^{-\beta J} - 1]/[e^{-\beta J} + 1]^2$. Moreover, there is a closed form expression for the average energy [26],

$$\frac{U}{NJ} = \frac{1}{2(1-\mu)}\left[ 1 - \frac{4\mu(3-\mu)}{4\sqrt{|\mu|} + \sqrt{(|\mu|+1)^3(3-|\mu|)}}\frac{2}{\pi}K(x)\right], \qquad (D.5)$$

where $\mu = 1 - 2\tanh(-\beta J/2)$, $K(x)$ is the complete elliptic integral of the first kind, and $x(\beta J) = [4\sqrt{|\mu|} - \sqrt{(|\mu|+1)^3(3-|\mu|)}]/[4\sqrt{|\mu|} + \sqrt{(|\mu|+1)^3(3-|\mu|)}]$. By inserting Eqs. (D.4) and (D.5) into Eq. (D.1), we thus obtain the entropy per particle for the triangular Ising lattice.

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
