# Peer review of "Localized dopant motion across the 2D Ising phase transition"

_SciPost Physics_

## Round 1 · Referee Report · Anonymous (Referee 1) · 2024-6-9

Strengths

The work provides nice results about an old problem (single mobile hole in a AFM or F background) in a different context (finite temperature).

Weaknesses

The paper does not discuss previous literature on the T=0 case. The role of quantum fluctuations (t-J model) may be compared to the role of thermal fluctuation.

Report

The paper presents solid results on the dynamics of a single hole in a 2D Ising spin background at finite temperatures. The calculations seem reliable and the results are interesting, predicting hole localization in all regimes, ferro/af, and high/low temperatures.

This problem restricted to T=0 has been investigated extensively in the late 80's and early 90's by many authors and I am surprised to see no reference/discussions. Clearly, the zero-temperature confining potential plays the same role at low temperatures (in the AF Ising case for example). Also, quantum fluctuations have been studied at T=0 ($t-J_z-J_\perp$ or $t-J$ models) and one may wonder whether thermal fluctuations may have a similar impact on the hole propagation by healing the spin defects introduced by the hole motion in the Néel phase.

The restriction of the hole motion to a line looks a bit artificial to me. One may wonder how the results depend on such a hypothesis.

Requested changes

Could the author consider my previous comments ?

Recommendation

Ask for minor revision

  • validity: high
  • significance: high
  • originality: good
  • clarity: good
  • formatting: good
  • grammar: good

Author:  Kristian Knakkergaard Nielsen  on 2024-06-10  [id 4550]

(in reply to Report 1 on 2024-06-09)
Category:
remark

Dear Referee,

Thank you for your diligent reading of the paper. I am happy to see that you find that the "...work provides nice results..." and that it "...presents solid results on the dynamics of a single hole in a 2D Ising spin background at finite temperatures...". Below, I briefly comment on the questions raised by the Referee.

Best regards, Kristian Knakkergaard

1) Lack of references to early papers. I fully acknowledge the lack of citations here. I will absolutely include such in my resubmission and also comment on the interesting and deep links to high-temperature superconductivity.

2) Thermal fluctuations versus flip-flop interactions. The idea that thermal fluctuations play a similar role to flip-flop spin interactions is certainly very interesting. In the present setup, however, I do not see a direct link. The reason is that the system I describe is actually closed, there is no coupling to an external heat bath. Therefore, there is no mechanism by which the thermal system can repair the "damage done" by the hole as it moves. Instead, I simply assume that the system starts out in the thermal Gibbs state of the spins, and then track the closed system dynamics as the hole is inserted. However, I really do think that this idea intriguing, and I will be sure to comment on it in my revised conclusions.

3) Restriction of hole motion to 1D I acknowledge that the restriction of the motion makes it hard to tell exactly what will happen if the hole is allowed to move in 2D as well. It still remains the case, however, that the hole experiences a fluctuating magnetic energy as it moves. Therefore, it may very well still be the case that the hole is localised. Special care is of course also needed here, because 2D is the marginal dimension for localisation. Moreover, as there are now several paths to the same point, there is no sense in which the motion here gives an effective potential of the hole only depending on its current position - it would generally depend on its entire path.

---

## Round 1 · Referee Report · Anonymous (Referee 2) · 2024-6-12

Report

In this work, the author studies the dynamics of a single hole in various types of 2D Ising spin lattices. In particular, the author considers the square and triangular lattices with ferromagnetic and antiferromagnetic couplings, both showing a finite temperature phase transition in the ferromagnetic case, whereas an antiferromagnetic phase can only appear on the square lattice. Importantly, the motion of the hole is confined to a single one dimensional line, allowing the author to perform efficient numerical simulations and formulate theoretical arguments for its long time dynamics. Under these conditions, the hole remains localized for all non-zero temperatures and Ising couplings, irrespective of whether the spin system undergoes a phase transition, though the mechanism of localization changes in different parameter regimes. The author analyzes the temperature and coupling dependence of the localization length in detail. As expected, the square and triangular lattices show qualitatively similar behavior in the ferromagnetic case. For antiferromagnetic couplings, the square lattice undergoes a phase transition to a Neel state, featuring pronounced coherent oscillations in the motion of the confined hole. Similar oscillations are washed out on the triangular lattice, due to the exponentially large ground state manifold stemming from frustration.

I find the paper well written and interesting, and I believe that the results deserve publication. However, I have certain concerns I would like the author to address, before I can recommend publication in SciPost Physics. In its present form, I am not convinced that the paper meets the acceptance criteria of SciPost Physics, but a revised version could be reconsidered.

In the paper it is emphasized at several points that finding localization in the ferromagnetic phase is an exciting and surprising result. In fact, I would naively think that this is simply an artifact of the motion of the hole being confined to a 1d line. There is a finite density of domain walls at any finite temperature even in the ferromagnetic case, some of those domain walls will inevitably intersect this 1d line, and the hole can not go around these obstacles due to its confinement to a 1d geometry. In this sense, the situation seems very similar to how Anderson localization takes place in arbitrarily small disorder in 1d.

By the reasoning above, I would expect that the localization studied in this paper is extremely fragile in the ferromagnetic regime, and the hole becomes delocalized as soon as a finite hopping in the perpendicular direction is introduced, or the spins become dynamical due to other terms beyond the classical Ising couplings. I do see value in testing and verifying the localization of the hole in this somewhat artificial one dimensional geometry, but I believe that the limitations of this analysis should be discussed more openly in the paper.

Due to these weaknesses, I do not believe that it is satisfyingly established that the paper opens a new pathway for research or a link between research areas. I think that for that, at the very least, a short discussion of which features are naively expected to remain robust in a more realistic / more isotropic geometry would be warranted, even if obtaining rigorous results is challenging.

A few other remarks:

- There is a typo in the first paragraph of the Introduction, “results are unfortunately limited … to short times and/or short times”, instead of short times and/or small system sizes, I presume.

- I am confused about the indices appearing in eq. (10), are they correct? I thought all y indices should be 0 here.

- A prefactor 1/tau seems to be missing in eq. (15).

- The inset of Fig. 3b is not very clear to me. I am confused about what the colored lines with arrows are supposed to show (these lines are rather hard to see, by the way). The blue one indeed naively seems consistent with a backscattering event, but I am less sure about the red one. I can not see any clear turn there, just fluctuations that look similar to the ones at many other points along the trajectory. How is this consistent with the statement about back-scattering, does this reflect sample-to-sample fluctuations?

- I am not sure I understand the argument about the non-monotonous temperature dependence of the localization length in Sec. 4.2. Why is the scaling of l_ave and l_fl with t/J relevant for this? I would have thought that non-monotonicity can arise when the slopes of these two length scales as a function of T have different signs at their crossing point.

- In Sec. 4.4, there is a statement about the localization length at the critical temperature being quantitatively close to its zero temperature limit on the antiferromagnetic side. I am wondering how justified this claim is, given the log scale in Fig. 7.

Recommendation

Ask for minor revision

  • validity: -
  • significance: -
  • originality: -
  • clarity: -
  • formatting: -
  • grammar: -

Author:  Kristian Knakkergaard Nielsen  on 2024-06-13  [id 4568]

(in reply to Report 2 on 2024-06-12)
Category:
answer to question
correction

Dear Referee, thank you for your detailed reading of my Article. I am thankful that you find "... the paper well written and interesting..." and that you "...believe that the results deserve publication". I appreciate your detailed comments, which I respond to below. With this, I hope that you can support publication.

Major remark: Robustness of results beyond 1D motion and Ising interaction. I agree with the referee that a discussion about the generalization of the model is warranted. I have, therefore, written a quite extensive discussion section (Sec. 5) that precisely goes into what might happen when the hole is allowed to move in 2D and other generalization of the model.

Briefly speaking, I think the referee is correct that when the hole moves in 2D, it will be able to propagate around domain islands of opposite spin. This suggests that the dopant should eventually be able to delocalize. Indeed, the model superficially seems very reminiscent of electron moving in a crystal lattice with localized defects that they can scatter on. By analogy, one should expect that the dopant can at least move diffusively with weak localization corrections. At high temperatures, however, the density of defects becomes on the order of unity, and the corresponding mean free path is on the order of the lattice spacing. This could suggest that the dopant remains localised in the high-temperature regime. Therefore, one can hypothesize that the observed reversed metal-insulator crossover seen in the present mixed-dimensional model is replaced by an actual transition, when the dopant is allowed to move in 2D.

I hope that the referee will take a look at the newly written Discussion section for further details on these matters.

Minor remarks.

1) I have corrected the typo related to "...short times and/or short time...". Thank you for spotting this.

2) The indices in Eq. (10) were indeed incorrect. They are now corrected.

3) Yes, a factor of $1/\tau$ was missing in Eq. (15).

4) I agree that Fig. 3 needed to be claried. I have done so now, and made the back-scattering picture more concrete. In particular, I now state that: "The physical picture is, hereby, that the dopant will eventually back-scatter off the emergent effective potential $V ( x)$ [inset of Fig. 3(b)]. Indeed, taking the enhancement factor of 2 into account, I check when $|V_{\sigma}(x)|$ first exceeds $\sqrt{2} t$ for each spin realisation $\sigma$ and average over the achieved mean-square distance $x^2$ over all the spin realisations. This quantitatively recovers the full numerical solution both in terms of the localisation length $l_{\rm loc}$, and in terms of the standard deviation around this value." I believe that this answers the referee's question, which I am happy that you brought up.

5) Non-monotonous temperature dependence. The central point is that at infinite temperature, $\beta |J| = 0$, the localization length scales as $(t/J)^2$. For any finite temperature, $beta |J| > 0$, however, the scaling goes as $f(\beta |J|) * (t/|J|)$. This means that when $|J|/t$ becomes low enough, there will be a situation where the larger scaling of $(t/J)^2$ at$ beta|J| = 0$ is above $f(\beta |J|) * (t/|J|)$ for some $\beta |J| > 0$. Since the hole eventually delocalizes in the ferromagnetic phase, the curve is non-monotonic for low enough$|J|/t$. I have now clarified this in the revised manuscript.

6) Critical temperature behavior vs. zero-temperature behavior at low $J/t$. They are quite similar, I would say. There is just a factor of 1.5 between them for lo $J/t$. This is now clarified in the revised manuscript.

---

## Round 2 · Referee Report · Anonymous (Referee 2) · 2024-6-26

Report

I appreciate the reply of the author, in particular, the extended discussion section that was added to the resubmitted manuscript. However, in light of this discussion, I can not recommend the publication of the paper in SciPost Physics. My main takeaway is that the physics explored in this work is too special to the somewhat artificial 1d geometry to offer much insight into the behavior of 2d systems. The extended discussion in Sec. 5 essentially just summarizes expectations that rely on what was already known about the localization of electrons in 1d and 2d lattices. I also do not see a clear way to extend the approach applied in this work beyond the special case of 1d hole motion. Moreover, the results presented here, while certainly valuable, essentially confirm the naive picture of localization in 1d for arbitrarily weak disorder (in the ferromagnetic case), or the presence of a linear confining potential (in the antiferromagnetic phase). For these reasons, I believe that the paper is better suited for publication in SciPost Physics Core.

Minor comments:

  • I appreciate the updated references, with more early works cited. Perhaps the author could expand this list even further (e.g. with Sachdev PRB 39, 12232, or Manousakis RevModPhys 63, 1).

  • I think that the fact that localization is actually the expected behavior in the 1d motion of a single particle for arbitrarily weak disorder (~domain wall density), on general grounds, should be explicitly mentioned in the introduction. The current phrasing is implying the opposite.

  • I believe that the discussion of the non-monotonous temperature dependence of the localization length around eq. (21) could still be made clearer. If I understand it correctly, l_ave was shown to diverge at infinite temperature. As it decreases with decreasing T, eventually it becomes smaller than l_fl (because of the different scaling with respect to t/J), at which point it becomes the relevant scale of localization. If this intersection happens where l_ave still had a positive slope with respect to T, this tendency has to turn at some T, because the hole becomes delocalized for T=0, so the localization length has to diverge in that limit. Is this correct?

Recommendation

Accept in alternative Journal (see Report)

  • validity: -
  • significance: -
  • originality: -
  • clarity: -
  • formatting: -
  • grammar: -

Author:  Kristian Knakkergaard Nielsen  on 2024-07-02  [id 4596]

(in reply to Report 1 on 2024-06-26)
Category:
remark
answer to question
reply to objection

Response to Reports 1 and 2

I appreciate the comments and objections from the Referees. I, however, do not agree that the current analysis does not inform us about the two-dimensional case, as the Referees imply. Indeed, if one immediately jumped to the 2D scenario and postulated an inverted metal-insulator transition, this would be hard to believe. But because of the detailed analysis in the current Article and the direct, solid evidence of an inverted crossover between these two phases for 1D motion, it is now very plausible that said transition should indeed occur when the hole can move in 2D.

Now please note again, that this is inverted from the usual metal-insulator case, where it is insulating at low temperatures and metallic at high temperatures. The mere existence of an inverted transition is highly nontrivial.

Along the same lines, I remain convinced that without such an analysis as the one performed here, one would not guess that the hole should localize at high temperatures even for 1D motion. It is only as a result of my current, thourough analysis that this becomes clear.

I, however, acknowledge that both Referees recommend publication in SciPost Physics Core, and I accept their recommendation.

Sincerely,

Kristian Knakkergaard Nielsen

Reply to minor comments from Report 1.

1) I thank the Referee for the references. I have included the one from Sachdev in the revised manuscript.

2) I agree with the Referee that the introduction should be slightly more carefully phrased. I have not included the sentence:

"While the hole in a ferromagnetic ground state will certainly delocalize, the same may not remain true as soon as one heats up the system by any infinitesimal amount. Indeed, when the system is not at zero temperature, occasional spin flips may act as local defects that generically has a localising effect in one dimension due Anderson localisation..."

I do not agree with the Referee, however, that one should expect localization in the current setup \emph{a priori}. Indeed, it is a core result of the paper that the spin flips do in fact work as local impurities for the hole motion.

3) I have made a final revision of the explanation of the non-monotonic behavior of the localization length. Please see the revised text for a full explanation.

---

## Round 2 · Referee Report · Anonymous (Referee 1) · 2024-6-27

Report

I share the opinion of the 1st referee: the setup of a hole moving along a line does not seem physical and does not give real insights into the true 2d physical problem. As a simple extension of previous work on ladder geometry, this work is more appropriate to the SciPost Physics Core.

Recommendation

Accept in alternative Journal (see Report)

---

## Round 2 · Author Response

I have now addressed all remarks and concerns from the referees. I thank them for their detailed reading of the manuscript, and I believe that this has helped to significantly improve the manuscript. The main remarks centred around a lack of citation to early work in the field and a lack of critical discussion of the results. I now provide both, and in particular extensively discuss and explain what may happen if the model is generalised beyond the mixed-dimensional setup with Ising interactions in the newly written Sec. 5.

---

## Round 2 · List of Changes

1) More extensively cite early papers dealing with the motion of dopants in quantum magnets (Introduction). 2) An extensive discussion (Sec. 5) of the results and how the investigated mixed-dimensional model may be generalised and the robustness of the discovered effects tested. This includes discussing: beyond 1D dopant motion, beyond the Ising model, increasing the doping level, connecting to an external bath. 3) Figure 3 and the associated text has been modified to more carefully explain the localisation mechanism at high temperatures. 4) Minor typos in equations (10) and (15) have been corrected.

---

## Editorial Decision

resubmitted